

# Greenland Ice Sheet late-season melt:
# Investigating multi-scale drivers of K-transect events

Thomas J. Ballinger[1], Thomas L. Mote[2], Kyle Mattingly[2], Angela C. Bliss[3], Edward Hanna[4], Dirk van As[5], Melissa Prieto[1], Saeideh Gharehchahi[1], Xavier Fettweis[6], Brice Noël[7], Paul C.J.P. Smeets[7], Carleen H. Reijmer[7], Mads H. Ribergaard[8], and John Cappelen[8]

*Correspondence to:* Thomas J. Ballinger (tballinger@txstate.edu)
[1]Department of Geography, Texas State University, San Marcos, TX, USA
[2]Department of Geography, University of Georgia, Athens, GA, USA
[3]College of Earth, Ocean, and Atmospheric Sciences, Oregon State University, Corvallis, OR, USA
[4]School of Geography and Lincoln Centre for Water and Planetary Health, University of Lincoln, Lincoln, UK
[5]Geological Survey of Denmark and Greenland, Copenhagen, Denmark
[6]Laboratory of Climatology, Department of Geography, University of Liège, Liège, Belgium
[7]Institute for Marine and Atmospheric Research, Utrecht University, Utrecht, the Netherlands
[8]Danish Meteorological Institute, Copenhagen, Denmark

Manuscript submitted to *The Cryosphere Discussions* on 19 December 2018, revised on 10 April 2019 and 15 July 2019

**Abstract**. One consequence of recent Arctic warming is an increased occurrence and longer seasonality of above-freezing air temperature episodes. There is significant disagreement in the literature concerning potential physical
connectivity between high-latitude open water duration proximate to the Greenland Ice Sheet (GrIS) and late-season (i.e. end-of-summer and autumn) GrIS melt events. Here, a new date of sea ice advance (DOA) product is used to determine the occurrence of Baffin Bay sea ice growth along Greenland's west coast for the 2011–2015 period. Over the two month period preceding the DOA, northwest Atlantic Ocean and atmospheric conditions are analyzed and linked to late-season melt events observed at a series of on-ice automatic weather stations (AWS) along the K-transect
in southwestern Greenland. Surrounding ice sheet, tundra, and coastal winds from Modèle Atmosphérique Régional (MAR) and Regional Atmospheric Climate Model (RACMO) provide high-resolution spatial context to AWS observations, and are analyzed along with ERA-Interim reanalysis fields to understand the meso-to-synoptic scale (thermo)dynamic drivers of the melt events. Results suggest that late-season melt events, which primarily occur in the ablation area, are strongly affected by ridging atmospheric circulation patterns that transport warm, moist air from
the sub-polar North Atlantic toward west Greenland. Increasing concentrations of North Atlantic water vapor are shown to be necessary to produce melt conditions as autumn progresses. While thermal conduction and advection off south Baffin Bay open waters impact coastal air temperatures, local marine air incursions are obstructed by barrier flows and persistent katabatic winds along the western GrIS margin.

## 1 Introduction

Substantial decline in Arctic sea ice extent and mass loss from the Greenland Ice Sheet (GrIS) have been observed for the last four decades (e.g. Serreze and Stroeve, 2015; Bamber et al., 2018). Under sustained climate warming, sea and land ice are becoming increasingly sensitive to changes in the frequency and duration of anomalous weather patterns (e.g. Overland and Wang, 2016; Hanna et al. 2014, 2018b). The overall mass balance of the GrIS has contributed roughly 0.5 mm year$^{-1}$ to global mean sea level rise since the early 1990s (van den Broeke et al., 2016),

with about 60% of the mass loss attributed to a decline in surface mass balance (SMB) and 40% associated with increased ice discharge (van den Broeke et al., 2017). Observed warming of near-surface ocean waters west of Greenland since at least the early 1990s is linked to accelerated submarine melt and outlet glacier retreat (Holland et al., 2008; Straneo and Heimbach, 2013) concurrent with more frequent summertime air temperature extremes along the coast (Hanna et al., 2012, Mernild et al., 2014). These findings raise the question of whether Baffin local ocean

conditions are of importance in governing the spatial extent and temporal variations in western GrIS melt.

       Conflicting evidence has been presented in literature over the past decade regarding the importance of the warming of the nearby ocean on GrIS surface melt. Regional climate model simulations have suggested that local sea surface temperature (SST) impacts on GrIS climate and SMB are negligible due to offshore flow arising from a prevailing katabatic wind regime (Hanna et al., 2014; Noël et al., 2014). Using a statistical approach, Rennermalm et

al. (2009) found contemporaneous Baffin open water (<15% sea ice concentration (SIC)) and western GrIS melt to be positively correlated in late summer (1979–2007). The authors noted that the strongest, statistically significant correlation (r=0.71) was observed in August near the K-transect (**Fig. 1**), and was attributed, in part, to wind-driven onshore transport of warm marine air. Hanna et al. (2009) evaluated lagged correlations between July coastal air temperatures at Ilulissat and Nuuk (~200 km north and ~300 km south of Kangerlussuaq, respectively) from Danish

Meteorological Institute (DMI) weather stations and adjacent, offshore HadISST1 SST values in the preceding and following 2 months. The authors noted a simultaneous, positive SST relationship with Ilulissat temperatures (r=0.56), while Nuuk temperatures were significantly correlated with offshore May–July SSTs (r>0.50) over the 1977–2006 record. Ballinger et al. (2018a) found significant interannual correlations (r>0.40, p≤0.05) during 1979–2014 between Baffin freeze onset dates (from the Markus et al., 2009 product) and September–December surface air temperatures

(SAT) at most DMI stations found along the west Greenland coastline. The authors found that significant, positive correlations between Baffin and Labrador SST and coastal SAT often persist through December after the onset of freeze. Applying a similar approach and melt/freeze dataset, Stroeve et al. (2017) showed Baffin and GrIS melt and freeze behaviors to be synchronous. The authors noted that years with anomalously early sea ice melt tended to have strong, upward turbulent heat fluxes and westerly winds atop developing open water that transported surplus heat and

moisture onto the ice sheet. Both studies indicated that the synoptic, upper-level circulation pattern is critical for modulating poleward heat and moisture transport and the surface warming/melt processes toward the end of the melt season. Ballinger et al. (2018a) proposed a sea ice-heat flux feedback whereby upward turbulent heat fluxes from Baffin Bay help maintain the high-pressure block aloft, with anticyclonic southerly winds both inhibiting the autumn/winter ice pack formation and transporting warm marine air onto the western Greenland coast. This potential

mechanism may have contributed to record Greenland Blocking events that occurred in October over successive years during the early-to mid-2000s (Hanna et al., 2018a).

A paucity of literature focused on the Baffin Bay open water influence on GrIS melt and SMB has left open the question of potential physical linkages. Our primary goal in this paper is to evaluate and determine whether the local ocean-atmosphere interactions have played a role in late-season GrIS melt events spanning the end of boreal

summer through autumn (e.g. Doyle et al., 2015; Stroeve et al., 2017). We posit that if Baffin Bay open water were to influence GrIS late-season melt events, the heat accumulated in the marine layer from ocean-to-atmosphere turbulent fluxes, would be transferred directly to the ice sheet by onshore westerly winds. In addressing this hypothesis, our analyses are geographically focused on the western slope of the GrIS with emphasis on the K-transect (**Fig. 1**) as this area – with its two on-ice AWS networks (described in Section 2.1) – is rich with in situ records relative

to the remainder of the ice sheet. We analyze data from these in situ sources and additionally from a global atmospheric reanalysis and regional climate models to address potential links between GrIS late season melt and the local-scale Baffin Bay marine layer for the period of overlapping data, 2011–2015. For completeness, we subsequently expand the scale of meteorological analyses to consider the influence of greater northwest Atlantic synoptic patterns on the melt episodes. The paper is organized in the following manner: Section 2 outlines data

sources, Section 3 describes methods employed, Section 4 covers the local and synoptic-scale atmospheric interactions with GrIS melt, Section 5 discusses key results, and Section 6 offers concluding remarks and makes suggestions for future research.

**2 Data**

**2.1 Passive microwave records**

Sea ice data are from the National Oceanic and Atmospheric Administration/National Snow and Ice Data Center (NSIDC) Climate Data Record of Passive Microwave SIC v3r1 product distributed by NSIDC (Meier et al., 2017). We use the "Goddard-merged" SICs that are produced using a combination of the NASA Team and Bootstrap algorithms applied to satellite passive microwave brightness temperatures (Peng et al., 2013). Daily observations are available over the 1979–2015 period at a 25 km by 25 km nominal grid spacing. The time series of daily SIC at each

grid cell were used to identify the sea ice date of advance (DOA), which is the date when SIC increases to 15% for the first time following the sea ice extent (SIE) minima after Steele et al. (2019). Regional mean DOAs for Baffin Bay were obtained from Bliss et al. (2019). The local DOA was determined from the time series of SICs between 1 October and 31 March at each grid cell, then the local mean DOA was computed from 13 grid cells within the domain 66.5 to 67.5°N and 53 to 55°W. SIE was computed by summing the area of grid cells (in km$^2$) where SIC$\geq$15%. At

the same spatial resolution as the SIC product, the passive microwave daily GrIS melt time series of Mote (2007, 2014) is also used to classify the ice surface environment in a binary manner (i.e. melt/no melt). Following the Ohmura and Reeh (1991) topographic regions, we assess GrIS melt conditions on the west-central portion of the ice sheet bounding the K-transect.

**2.2 AWS data**

Meteorological conditions from two AWS networks, the Programme for Monitoring of the Greenland Ice Sheet (PROMICE; stations prefix "KAN") and Utrecht University Institute for Marine and Atmospheric Research

(hereafter IMAU; stations prefix "S"), are used in this study (**Fig. 1**). To examine the possible Baffin Bay open water influence on observed surface air temperatures from the approximate terminus position nearest the ocean upslope across the longitudinal extent of the K-transect (referenced herein as the combination of PROMICE and IMAU stations), we conduct analyses based on temperature conditions monitored at KAN_B (see Section 3 for details). The operational record of KAN_B began in 2011 and represents the starting year for analysis, while the DOA record concludes in 2015 marking the end of our study period. Data are obtained and analyzed from seven weather stations distributed across the K-transect at ~67°N during this period. The transect spans an area from low-elevation tundra, approximately 1 km inland from the ice sheet glacier terminus (KAN_B), to the lower accumulation area (~1800 m) at >140 km from Russell Glacier terminus (KAN_U; see **Fig. 1** & **Table 1**). AWS data used here are recorded at approximately 2–3 m above the surface, though the heights of the air temperature sensors are known to fluctuate due to snow accumulation and melt season ablation (Charalampidis et al., 2015). Daily mean air temperature (°C), wind speed (ms$^{-1}$), and wind direction (0–360°) are obtained from IMAU and PROMICE station networks. The data series are mostly complete for the study period, and the few missing values are filtered out prior to analyses. Additional details on the respective PROMICE and IMAU AWS programs can be found in van As et al. (2011) and Smeets et al. (2018). We supplement K-transect data with daily mean DMI AWS surface air temperatures obtained from Sisimiut (WMO code 4234) and Kangerlussuaq (WMO code 4321). These data are used to further evaluate spatial links between local Baffin Bay open water and air temperatures from the tundra regions below the K-transect (Cappelen, 2018, 2019; **Fig. 1**).

**2.3 Atmospheric reanalysis and regional climate model fields**

A number of ERA-Interim reanalysis (Dee et al., 2011) variables are analyzed at their native 80-km spatial resolution to assess atmospheric conditions across Greenland and the northwestern Atlantic Arctic sector. ERA-Interim surface and upper-air temperatures, wind speeds, and moisture conditions exhibit relatively small biases compared to Arctic observations (Bromwich et al., 2016). Regional climate models MAR and RACMO are also forced with ERA-Interim fields at their lateral boundaries. Tropospheric winds and specific humidity from the 1000 to 200 hPa (1000-200) atmospheric layer are used to calculate a moisture flux variable referred to as integrated water vapor transport (IVT); IVT is then classified using a self-organizing map (SOM) approach (see formal description in the Methods section below). Surface-atmosphere interactions and regional atmospheric circulation characteristics are further evaluated using 80-km native resolution ERA-Interim fields including mean sea-level pressure (MSLP), 10-meter (10-m) winds, latent and sensible heat fluxes, 500 hPa geopotential heights (GPH), and winds averaged over the 1000-700 hPa layer.

Wind speed and direction at 10-m and 850 hPa from MAR and RACMO are evaluated to understand low-level atmospheric flow over ocean-land-ice sheet areas surrounding the K-transect. Secondarily, we briefly discuss inter-model differences within the planetary boundary layer and biases against AWS observations. Both regional climate models are specifically developed for simulating polar weather and climate, in particular over the Greenland ice sheet (e.g., Fettweis et al., 2011; Noël et al., 2018). MAR v3.9 fields at 15 km are used here (see Fettweis et al. 2017 for a detailed model description). Relative to MAR v3.8 used in Delhasse et al. (2018), the main changes to MAR v3.9 consist of enhanced computational efficiency, adjustments to some of the snow model parameters to better

compare with in situ observations, and improved MAR dynamical stability by increasing the atmospheric filtering. RACMO2.3p2 fields (hereafter referenced as RACMO2) at a horizontal resolution of 5.5 km are also used (Noël et al., 2018). Model physics have not changed relative to the previous 11 km version described in Noël et al. (2018). The refined spatial resolution of the host model improves the depiction of topographically complex terrain at the GrIS margins, such as small peripheral glaciers and ice caps, and the representation of near-surface, local winds.

**2.4 North Atlantic atmospheric indices**

Daily atmospheric indices are examined to characterize near-surface and upper-level conditions within a historical context. The Greenland Blocking Index (GBI) (Hanna et al., 2018) describes daily mean 500 hPa GPH values from 60-80°N and 20-80°W. The North Atlantic Oscillation (NAO) index used here is adapted from Cropper et al. (2015) and represents station-based daily MSLP differences between Iceland and the Azores. Both versions of the respective indices are normalized by their day of year means and standard deviations for the common 1951–2000 base period.

**3 Methods**

Above-freezing daily air temperatures at on-ice AWS locations represent an indicator of ice-sheet melt (Hock 2005). A composite approach is applied to characterize atmospheric conditions underlying late-season GrIS melt events, defined here as occurring at the conclusion of boreal summer (i.e. late August) and during autumn preceding sea ice advance on Baffin Bay (**Table S1**). Two constraints are placed on the composite analyses. The first constraint involves the length of the analysis period prior to DOA. Baffin Bay ice freeze onset dates and sea surface temperatures have shown a two month lead time and positive, significant correlation with west Greenland coastal air temperature variations (Hanna et al., 2009; Ballinger et al., 2018). Composites are constructed over a similar two-month (60-day) period with data additionally sub-divided into 60-31 day (i.e. [-60,-31]) and 30-1 day (i.e. [-30,-1]) bins preceding each Baffin Bay DOA from 2011 to 2015. Regarding the second constraint, the composite technique is intended to isolate meteorological processes most common during KAN_B daily mean air temperature events of ≥0°C (T+) versus <0°C (T-) and the spatial consistency of such conditions across the longitudinal extent of the transect. To resolve the spatial cohesiveness of melt events along the K-transect, daily mean air temperatures at each K-transect station are composited (i.e. averaged) for KAN_B T+ and T- days in the [-60,-31] and [-30,-1] periods. On average during 2011-2015, KAN_B T+ events characterize ~46% of days in the [-60,-31] period, while [-30,-1] events are less frequent and occur 9% of days preceding Baffin Bay DOA. T+ comparisons between KAN_B and other K-transect stations are provided in **Table 2**. Temporal overlap between above-freezing temperatures at KAN_B versus S5 (77%) and KAN_L (46%) in columns 2-3 of **Table 2** suggest spatial coherence in the physical mechanisms forcing melt at least across part of the lower ablation area.

In a similar fashion to Carr et al. (2017), a Wilcoxon test is used to evaluate differences in atmospheric variables and indices between T+ versus T- events. This nonparametric test is intended for continuous data series that do not follow underlying assumptions of the normal distribution making it appropriate for comparative analyses between extreme and non-extreme meteorological observations. The null hypothesis of no difference in atmospheric conditions between cases is rejected at the 95% confidence level when p≤0.05.

To classify synoptic patterns of atmospheric moisture transport about Greenland, integrated water vapor transport (IVT) is calculated from ERA-Interim data following Eq. (1):

$$IVT = \frac{1}{g} \int_{1000 \text{ hPa}}^{200 \text{ hPa}} qV \, dp \tag{1}$$

where g is gravitational acceleration (9.80665 m s$^{-2}$), q is specific humidity (g kg$^{-1}$), V is the vector wind (m s$^{-1}$), and dp represents the difference between atmospheric pressure levels.  Pressure levels are spaced at 50 hPa intervals

between 1000 hPa and 500 hPa, and 100 hPa intervals from 500 hPa to 200 hPa.  To control for the IVT seasonal cycle, the percentile rank of IVT (IVT PR) is calculated by comparing 6-hourly IVT values at each grid point to the distribution of all IVT values within a 31-day centered window at that grid point during 1980-2016.  The four 6-hourly IVT PR values during each day are then averaged to generate daily mean IVT PR values.

       Daily mean IVT PR data are classified using the SOM technique to produce a matrix of moisture transport

patterns, or nodes, that typically occur over the Greenland region.  The SOM is based on an unsupervised machine learning algorithm that classifies each daily IVT PR field into the closest matching SOM node.  As in Mattingly et al. (2016), we utilize a 5x4 SOM configuration.  We further classify SOM nodes—based on visual inspection of IVT PR patterns over Greenland—into "wet" (anomalously high IVT PR over Greenland), "neutral" (near climatological median IVT PR), and dry (anomalously low IVT PR) SOM node groups, and test whether the frequency of IVT

patterns falling into each node group differs across T+ and T- events.

**4 Results**

**4.1 Characteristics of Baffin Bay and west Greenland late-season melt**

       Time series depicting the conclusion of melt conditions across Baffin Bay and the western GrIS are shown in **Fig. 2**.  Later sea ice formation is apparent in the DOA series, particularly from around 2000.  Similarly, the start

of the last ≥3-day sequence of Region 3 (west-central Greenland; see inset in **Fig. 1**) 2% or 4% melt area also suggests progressively later melt (and a later onset of freezing conditions).  A significant break in the 2% series is highlighted by a drastic increase in variability from 1979–1999 (σ=21.17) to 2000–2015 (σ=44.25) that is also present in the annual discharge records from the nearby Watson River and Tasersiaq ice sheet catchments (Ahlstrøm et al., 2017; van As et al., 2018).  Differences between the beginning of Baffin Bay sea ice advance and the end of the ice sheet

melt season have clearly narrowed in part due to regional melt season lengthening.  Some GrIS melt events since 2000 have notably occurred after seasonal sea ice formation (i.e. 2002, 2004–2005, 2010).

       Relative to the climatology defined as 1981–2010, sea ice advanced ~11d earlier in 2011 and on average ~6d later in 2012–2015 (**Table S1**).  Inspection of the DOA for individual grid cells adjacent to the Sisimiut AWS, which is labeled 4234 in **Fig. 1** and located ~150 km west of the K-transect ice sheet margin, reveals a northward-extending

notch where ice forms ~30–60d later than the Baffin-wide DOA (**Fig 3**).  Interannual differences in the region's ice cover advance, such as those observed 2011-2015, often depend on physical factors including regional winds, ocean heat transport, and water-mass changes (Myers et al., 2009; Ribergaard, 2014).  For instance, strong offshore winds and poleward circulation of warm water from the West Greenland Slope Current often contribute to the local open water persistence, while southward Arctic Water transports support earlier ice formation patterns found in the east and

north (Curry et al., 2014).

**4.2 Local meteorology of melt versus non-melt cases**

Composites of air temperature, wind speed and direction by KAN_B T+ and T- events are shown in **Fig. 4**. Across the transect, composite air temperature differences (T+ minus T- events) are warmer by roughly 7 to 8°C in the [-60,-31] window and 12 to 13°C in the [-30,-1] period (**Fig. 4a**). These differences tend to be smallest near the coast and increase to S6 in the mid-ablation area (~1000 m asl) where contemporaneous melt occurs ~14–16% of the time (**Table 2**).

KAN_B receives stronger, more southerly winds during T+ events that tend to become weaker and more southerly in autumn (i.e. [-30,-1]). There tends to be a general wind speed increase during T+ events, while direction remains more or less unaltered. Statistically significant directional change is more southerly, which is evident above the long-term equilibrium line altitude at S9. These katabatic winds are deflected to the right (southeasterly) by the Coriolis force as they travel toward Baffin Bay (van den Broeke et al., 2009) and may be aided additionally by synoptically-driven southerly winds. There is also evidence of wind speed intensification during [-60,-31] T+ events (~1–2 m s$^{-1}$) and to a lesser magnitude similar relationships hold during the [-30, -1] period (**Fig. 4b**). Increased wind speeds in T+ versus T- events are likely affected by the seaward enhancement of the pressure gradient aided by longer periods of nearshore open water (**Fig. 5**), while increased synoptic cyclone activity and lower MSLP over Baffin Bay during the summer-autumn season transition may also enhance offshore flows (McLeod and Mote, 2015). Wind speed increases may also initiate positive (downward) sensible heat fluxes associated with low elevation ice melt (**Fig. 6**). Just offshore, sensible and latent heat fluxes are generally negative (upward) in T- events and near zero to slightly positive during melt occurrences; positive turbulent flux differences in T+ versus T- events are apparent over oceanic areas from the Labrador Sea extending northward into Baffin Bay and the western edge of Greenland. This indicates decreased heat transfer from the ocean to atmosphere during T+ events relative to T- conditions, with southerly near-surface winds indicative of warm air advection (**Fig. 6**).

The K-transect offshore flows appear to represent a dynamic barrier to Baffin Bay marine layer intrusions. To further examine lower tropospheric flow across the ocean-land-ice interface, we similarly composite MAR and RACMO2 winds at 10-m and 850 hPa (**Figs. 7** and **8**). In general, the MAR and RACMO2 winds show directional consistency with overlaid PROMICE and IMAU winds with a slight southerly bias in both products for [-30,-1] at KAN_M and S9 in the upper ablation area. Simulated wind speeds are ~20-50% stronger during T+ versus T- events (not shown) as corroborated by most of the AWS observations (**Fig. 4b**), helping enhance sensible heat flux into the ice sheet in areas where near-surface temperatures are above freezing. T+ events in RACMO2 and MAR generally capture AWS observed wind speeds in the upper ablation area at KAN_M (r$^2$≥0.77) and lower accumulation area at KAN_U (r$^2$≥0.77) with low root mean squared errors (RMSE<1.50 m s$^{-1}$ in all cases). A slight, positive bias in model-derived wind speed is evident at the ice sheet edge near KAN_B (MAR r$^2$ = 0.33, RACMO2 r$^2$ = 0.58). We note that height and therefore surface roughness differences between the AWS measurements (2–3 m) and regional model 10-m winds may explain a portion of the bias at KAN_B. The 10-m winds extending from the west Greenland tundra into eastern Baffin Bay are notably weak and northerly in T- events while T+ events are comparably stronger and southerly; **Figs. 7** and **8**).

**4.3 The role of North Atlantic atmospheric patterns**

Overplots of 500 hPa GPH, 1000-700 hPa mean wind, and IVT for Greenland and the surrounding northwest Atlantic region are shown in **Fig. 9**. Whereas T- events tend to be characterized by northerly winds over the 1000-700 hPa layer, the T+ events indicate southerly, on-ice transfer of subpolar air aided by the presence of an upper-level trough over Baffin Island and downwind ridging over Greenland (see left versus right panel plots in **Fig. 9**, respectively). Higher GPH values are found over the ice sheet during T+ events as the 540 dam (i.e. 5400 m) contour extends across central Greenland, while the same contour is located south of the island in T- events. In both T+ cases, 1000-700 winds circulate poleward over the north Labrador Sea aiding the heat and moisture transfer (as shown by heightened IVT values in T+ relative to T-) to the western Greenland ice sheet during [-60,-31] and [-30,-1] (**Fig. 9**). One notable difference between T+ cases is that low-level southerly winds originate deeper (~50°N) in the North Atlantic during the T+ [-30,-1] case. Local IVT maxima in both T+ events are concentrated over the southwest tip of the island but remain ~100 kg m$^{-1}$ s$^{-1}$ near the K-transect. Comparatively, the depth integrated moisture flux over much of the west coast increases by a factor of 2–3 (4–5) during [-60,-31] ([-30,-1]) T+ versus T- events. This finding suggests that moist, onshore flow drives late season GrIS melt events, and moreover higher IVT is necessary to produce melt conditions as autumn progresses.

To further characterize and differentiate weather conditions by T± event, we utilize a SOM of IVT PR fields and composite SOM-classified daily IVT wet, dry, and neutral patterns (**Fig. 10a, b**). Analyses of the aggregated frequencies suggest that the wet patterns (with anomalously high IVT versus climatology) occur significantly more often in T+ versus T- events, and such nodes are more common by a factor of >4.5 in the [-30,-1] period. (**Fig. 10b**). While some caution should be exercised as the absolute frequency of these patterns decreases from roughly early ([-60,-31]) to late ([-30,-1]) autumn, humid atmospheric conditions appear to cause late-season melt. Increased incidence of wet patterns coincides with negative (positive) NAO (GBI) (both >|0.50| in T+ events; **Fig. 10c**), and a synoptic environment characterized by high surface (upper-level) pressure anomalies. This is confirmed by **Fig. 9** whereby the higher 500 GPH values and 1000-700 mean winds during T+ events transport warm and moist air masses from Labrador Sea and southerly maritime latitudes to much of the western slope of the ice sheet to facilitate ablation-area melt.

**5 Discussion**

West Greenland summer and autumn air temperature variability and trends during the last 3–4 decades have shown strong response to increased frequency and intensity of Greenland high-pressure blocking, negative NAO patterns, and positive North Atlantic SST anomalies aided by background anthropogenic forcing (Hanna et al., 2016; McLeod and Mote, 2016; Ballinger et al., 2018a; Graeter et al., 2018). The current North Atlantic "warm period" since the mid-1990s is characterized by a positive Atlantic Multidecadal Oscillation phase and rising SSTs around southwestern Greenland (Myers et al., 2009; Ribergaard, 2014), including just offshore of Sisimiut (WMO code 4234 in **Fig. 1**) (orthogonal trend = +0.03°C year$^{-1}$, p<0.05, for 1995-2015 period using the SST product described in Ballinger et al., 2018b). Warming waters around the island are influencing Baffin Bay sea ice and west GrIS melt processes (Hanna et al., 2013; McLeod and Mote, 2015; Ballinger et al., 2018a) and seasonality toward earlier melt and later freeze (Stroeve et al., 2017). This melt area about K-transect suggests local low SIC and open water, inferred upward turbulent atmospheric heating, and onshore winds could influence nearby terrestrial melt events. Moreover,

Sisimiut SSTs fluctuate with air temperatures (over the 60 days preceding DOA) in a statistically significant fashion for 2013-2015 at most K-transect stations with some distance decay noted upslope from the edge of the ablation area at S9 (**Fig. S1**).  Summer and autumn west Greenland near-coastal air temperatures are modulated by the thermal properties of bordering SSTs (Hanna et al., 2009; Ballinger et al., 2018a).  Interannual differences in the strength of SST-air temperature relationships (i.e. 2013–2015 versus 2011–2012) suggest: 1) processes driving warming ocean

waters and air temperatures over the GrIS are independent when disparate wind directions occur at or near the ocean-tundra-ice sheet boundaries in years of weak-to-zero correlation (e.g. katabatic flows contrasting near-coastal barrier flows (van den Broeke and Gallée, 1996)), or alternatively 2) large-scale atmospheric circulation (i.e. near-surface and upper-level meridional winds) during years of positive, statistically significant correlations, modulates the near-shore surface open water and ice sheet air temperatures (Stroeve et al., 2017).  We recognize that synoptic patterns

may not necessarily be mutually exclusive in these examples, but the study objectives do not include comparison of high and low pressure features around Greenland for specific melt and non-melt events.

A number of studies have suggested that Baffin Bay marine layer interaction with the ice sheet boundary layer is obstructed by zonal and meridional flows such as the west coast plateau jet feature and katabatic winds (Hanna et al., 2009; Moore et al., 2013; Noël et al., 2014).  Moore et al. (2013) noted a directionally consistent southerly 10-

m wind field extending over the western half of Greenland in summer and winter, while observational studies similarly indicate a high frequency of southerly-to-southeasterly winds over the K-transect (van den Broeke et al., 2009).  Southerly (easterly) 10-m winds are strongly linked to melt across two thirds (the southern third) of the ice sheet (Cullather and Nowicki, 2018).  For an expanded spatial perspective, we briefly examine air temperatures at the next PROMICE station installment approximately 700 km north at Upernavik (UPE; 72.89°N).  We find UPE_L (220 m

asl on the ice sheet) melt occurs the day of KAN_B T+ events on >50% of occasions in both [-60,-31] and [-30,-1] windows (not shown). This suggests that above-freezing near-surface air often penetrates at least to PROMICE station UPE_L with a relatively warm air mass engulfing much of the west coast.  Our observational and regional model analyses further show that homogenous low-level winds extend coastward at least to the tundra-ice sheet interface near KAN_B (see **Figs. 4c**, **7**, and **8**) and produce a "blocking effect" that inhibits the inland penetration of near-

surface air from Baffin Bay (Noël et al., 2014).  Of note, the katabatic mechanism becomes stronger as Baffin Bay DOA approaches in late autumn, with more pronounced radiational cooling over the ice sheet further supporting winds that prevent incursions of local marine air (van As et al., 2014).

If late season K-transect melt is minimally influenced by local Baffin Bay open water, then what physical mechanisms drive the melt events?  Composites of the AWS K-transect observations and complementary regional

model output indicate that recent late-season melt events tend to be driven by southerly synoptic patterns as opposed to local marine forcing.  We provide evidence of this physical forcing by [-60,-31] and [-30,-1] T+ composites that show southerly flows of more warm, moist maritime air of lower latitude origins relative to T- cases.  As shown in **Fig. 9**, during the former event period, air is transferred off northern Labrador Sea to the west coast, while a path of more southerly flow directs moist North Atlantic air masses onto Baffin Bay and southwestern Greenland in the period

immediately preceding DOA.  These "wet" synoptic patterns occur frequently under anomalous (>|1σ|) positive GBI and negative NAO values (**Fig. 10b**, **c**).  We surmise from Mattingly et al. (2018) that such patterns are particularly

moisture-rich (≥85[th] percentile IVT climatological values) and often accompanied by atmospheric rivers impacting Greenland, and their occurrence causes ablation area melt in non-summer, low insolation months through cloud radiative effects (i.e. increased downward longwave radiation transfer into the ice surface), condensational latent heat release, increased near-surface winds and turbulent heat flux, and liquid precipitation (Doyle et al., 2015; Binder et al., 2017; Oltmanns et al., 2019). The southerly winds that propagate moisture northward off the northwestern Atlantic Ocean are a product of amplified upper-level geopotential height patterns and meridional winds in T+ versus T- events extending from Denmark Strait and Irminger Sea on the east coast of Greenland onto the ice sheet (**Fig. 9**). Mid-tropospheric ridging, which is more pronounced in [-30,-1] than [-60,-31] events, supports southerly winds that funnel heat and moisture from likely deeper in the Atlantic basin to Baffin Bay and southwest Greenland to stimulate sea ice and GrIS ablation area melt conditions (Ahlstrøm et al., 2017; Ballinger et al., 2018a,b; Hanna et al., 2018). Cullather and Nowicki (2018) similarly find collocated, positive MSLP and 500 hPa GPH anomalies over Denmark Strait and Irminger Sea tend to be associated with melt events in the basin encompassing the K-transect. Our analyses suggest that ocean-to-atmosphere turbulent fluxes are suppressed over Baffin Bay during T+ events, presumably due to the synoptic-scale warm air advection regime reducing the temperature and humidity gradients between the sea surface and atmosphere (Aemisegger and Papritz, 2018). Results further support North Atlantic-air-ice sheet coupling, rather than localized Baffin Bay ocean-atmosphere processes, as a strong driver of transition season melt before sea ice advances south of the K-transect. Synoptic patterns associated with negative summer NAO and positive GBI incidence strongly influence these melt events (**Fig. 10a, b, c**), prompting decreases in SMB, and increase in the K-transect equilibrium line altitude over the last 10-15 years (Hanna et al., 2013; Smeets et al., 2018).

**6 Conclusions**

Temporal co-variability between GrIS and Arctic sea ice mass loss suggests a possible feedback whereby adjacent open water conditions, ocean-to-atmosphere heat flux, and on-ice winds affect inland melt during the end of the melt season. Based on our 2011–2015 analyses bridging the end-of-summer/early autumn melt to the date of first-year Baffin Bay sea ice advance, we do not find evidence to support the hypothesis that local open water, resultant turbulent heating, and onshore winds have a pronounced impact on inland ice melt events. These thermodynamic processes, in particular, directly influence coastal air temperatures and have a fingerprint on marine outlet glacier behaviors (Carr et al., 2017), but are shown here to be inhibited by topographically-influenced flows and synoptic patterns whose interactions are not mutually exclusive. Furthermore, Baffin Bay warming coupled with a longer autumn open water period has been hypothesized to stimulate and invigorate upper-level, high-pressure blocking that promotes southerly air advection over the west Greenland coast (Ballinger et al., 2018a). This is consistent with the main conclusions of Noël et al. (2014), which suggest that while warming waters around Greenland minimally affect SMB beyond enhancing tidewater glacier retreat rates SST forcing may indirectly influence GrIS SMB changes through impacts on atmospheric circulation. However, in terms of direct forcing by the local marine layer, beyond a near-coastal influence, our AWS, regional, and synoptic wind analyses suggest that Baffin Bay does not represent a substantial advective heat and moisture source to the ice sheet during our 5-year analyses.

Future late season analyses, perhaps reconstructing K-transect meteorological conditions back to the origins of the modern sea ice record, might be insightful in comparing local ocean-ice sheet interactions spanning the 1990s

shift from colder to warmer Baffin Bay summer SSTs (Ballinger et al., 2018a). Assessing the temperature and pressure gradients and their vertical profiles derived from retrospective analyses would also be useful to categorize the structure, magnitude, and direction of regional winds, including the katabatic regime, in attempting to provide a longer-term perspective of analyses presented in this paper. Noël et al. (2014) hypothesized that future sea-surface warming may exacerbate the division between the local ocean and ice sheet by intensifying the temperature and pressure gradient and hence resulting katabatic winds. Baffin Bay climate and cryospheric changes in the last two decades suggest such an increased "blocking" mechanism may already be underway. Moreover, stronger katabatic winds might be enhanced further by an increasing intensity of autumn mid-troposphere high-pressure over Greenland (Hanna et al., 2018a, see their Fig. 1e). Synergistic future research should continue to monitor the spatial extent, drivers, and physical effects of late-season melt through observational products and regional modeling tools, including quantification of late season K-transect mass loss and runoff through the Watson River, contributions to subsurface/firn processes, and preconditioning effects on the following year's melt season.

*Author Contributions.* TJB and TLM conceived the study. TJB analyzed the observational data, with assistance from MP and SG, and led the writing of the manuscript. TLM developed and processed the satellite-derived GrIS melt data. KSM conducted the IVT classification and assisted with the creation of several figures. ACB developed the DOA series and contributed related figures. EH provided the daily GBI series, and DvA, CHR, PCJPS, MHR, and JC provided AWS or oceanographic data and support. BN and XF developed and processed regional model wind fields for RACMO2.3p2 and MAR v3.9, respectively. All authors provided valuable insights, feedback, and editing on manuscript drafts.

*Acknowledgements.* The KAN PROMICE weather stations are funded by the Greenland Analogue Project, and the IMAU K-transect stations are funded by the Netherland Institute for Scientific Research and its Netherlands Polar Programme. TJB acknowledges support from Texas State University, and IASC, NSF, and UAF for sponsoring and facilitating an APECS travel grant to attend the POLAR 2018 Open Science Conference where valuable project-related feedback was received. BN acknowledges funding from the Polar Program of the Netherlands Organization for Scientific Research (NWO) and the Netherlands Earth System Science Centre (NESSC). The authors thank David Bromwich and Jeffrey Miller for constructive comments on early results, and Thomas Cropper for making available his daily NAO series. Constructive remarks from Charalampos Charalampidis and two anonymous references were helpful in guiding manuscript improvements.

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

**Tables**

| AWS Station | Network | Latitude (°N) | Longitude (°W) | Elevation (m asl) | Distance to/from terminus (km) |
|---|---|---|---|---|---|
| KAN_B | PROMICE | 67.13 | 50.18 | 350 | 1 |
| S5 | IMAU | 67.08 | 50.10 | 500 | 6 |
| KAN_L | PROMICE | 67.10 | 49.95 | 670 | 12 |
| S6 | IMAU | 67.07 | 49.38 | 1000 | 37 |
| KAN_M | PROMICE | 67.07 | 48.84 | 1270 | 61 |
| S9 | IMAU | 67.05 | 48.22 | 1500 | 88 |
| KAN_U | PROMICE | 67.00 | 47.03 | 1840 | 142 |

**Table 1.** Summary details of the PROMICE and IMAU AWS stations utilized in this study, including their approximate geographic position (in decimal degrees), elevation, and distance from the ice sheet terminus moving west to east. KAN_B is located on the tundra, roughly 1 km to the west of the terminus. Distances are rounded to the nearest km as on-ice AWS sites are known to move ~50 - 150 m year$^{-1}$ (van de Wal et al., 2015).


| AWS T Compare | T+ n[-60,-31] | T+ %[-60,-31] | T+ n[-30,-1] | T+ %[-30,-1] | T+ %[-60,-1] |
|---|---|---|---|---|---|
| S5 vs KAN_B | 53 | 77 | 9 | 69 | 76 |
| KAN_L vs KAN_B | 32 | 46 | 8 | 62 | 49 |
| S6 vs KAN_B | 10 | 14 | 3 | 23 | 16 |
| KAN_M vs KAN_B | 2 | 3 | 1 | 8 | 4 |
| S9 vs KAN_B | 1 | 1 | 0 | - | 1 |
| KAN_U vs KAN_B | 0 | - | 0 | - | - |
| ∑ KAN_B T+ events | 69 | - | 13 | - | - |

**Table 2.** Counts of above-freezing daily mean air temperature (T+) events (n), 2011-2015, and the percentage of contemporaneous overlap (%) between T+ events at KAN_B and S5, KAN_L, S6, KAN_M, S9, or KAN_U. The [-60,-31] and [-30,-1] periods reference time windows before respective annual dates of Baffin Bay sea ice advance (DOA). As an example, 77% of the time in the 30 to 60-day (i.e. [-60,-31]) window preceding Baffin DOA the T+ air temperature threshold at KAN_B is also observed at S5.

**Figures**

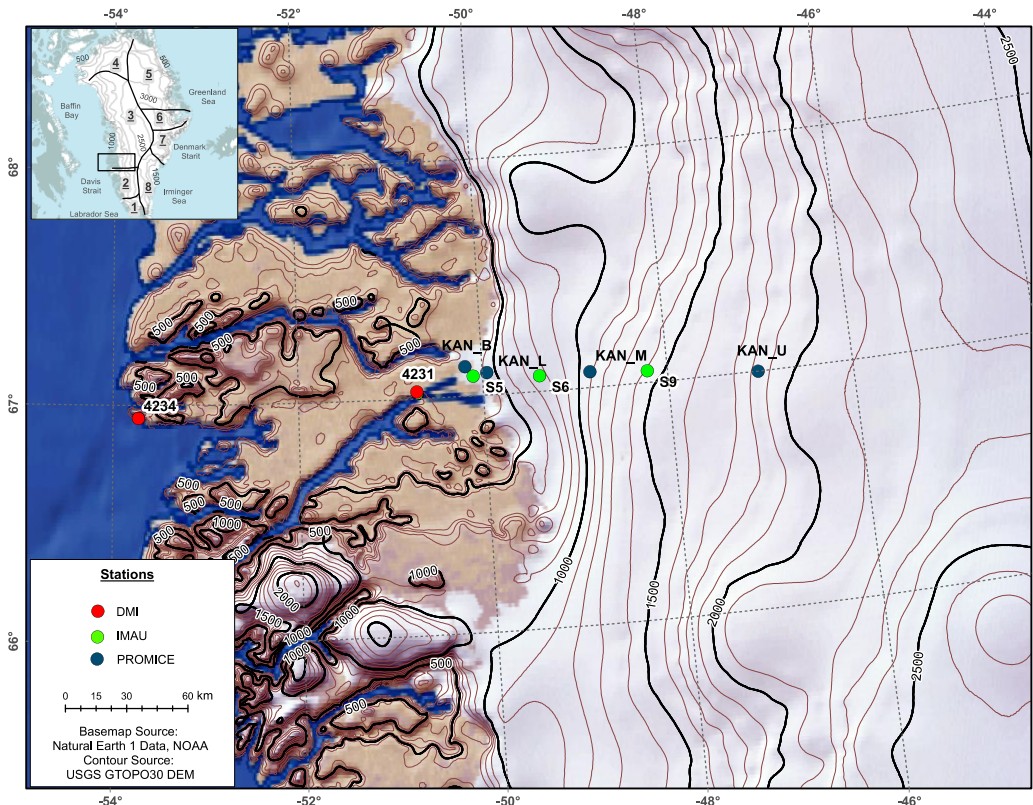

**Figure 1.** Study area map with PROMICE and IMAU K-transect sites and adjacent terrestrial DMI stations (Kangerlussuaq (WMO code 4231) and Sisimiut (WMO code 4234)). The inset displays the northwest Atlantic Arctic region with superimposed GrIS topographically-defined boundaries (adopted from Ohmura and Reeh (1991)).

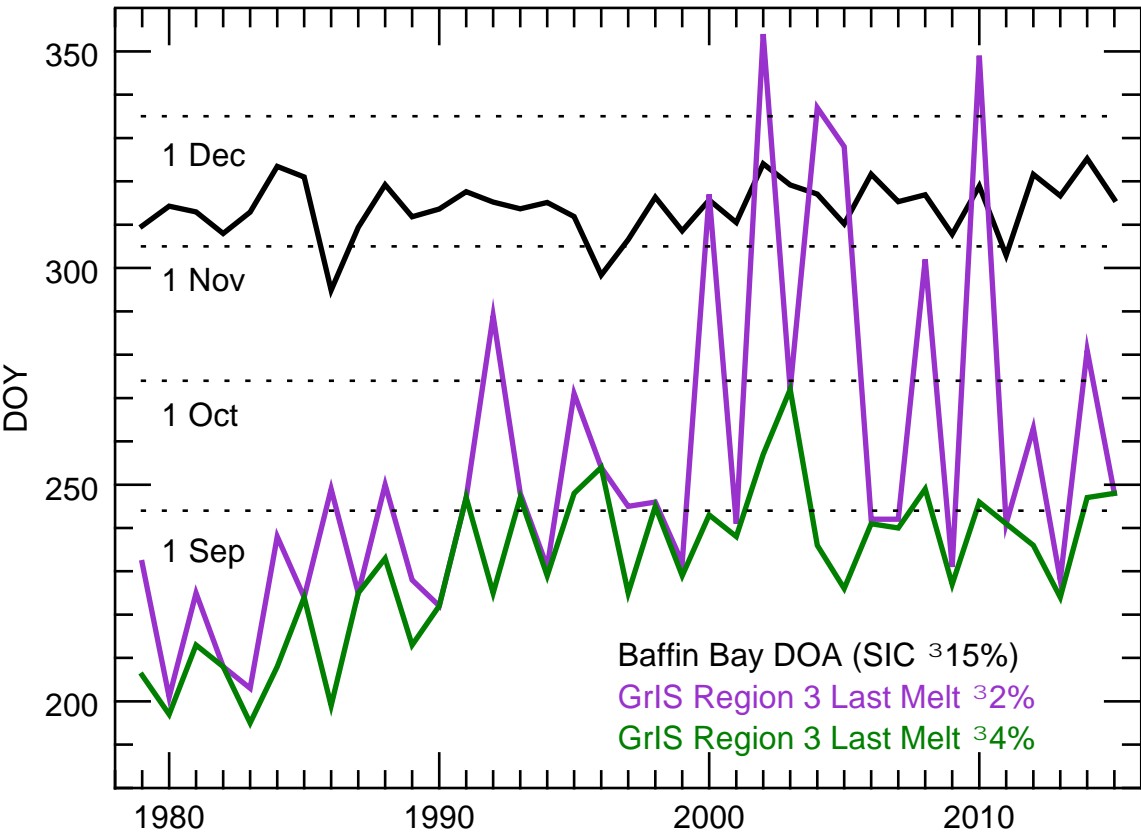

**Figure 2**. Passive microwave-derived time series, 1979-2015, of the Baffin Bay sea ice date of advance (DOA) and the date marking the beginning of the last 3-day period of at least 2% and 4% melt over Region 3 (see inset in **Fig. 1**) of the Greenland Ice Sheet (GrIS).

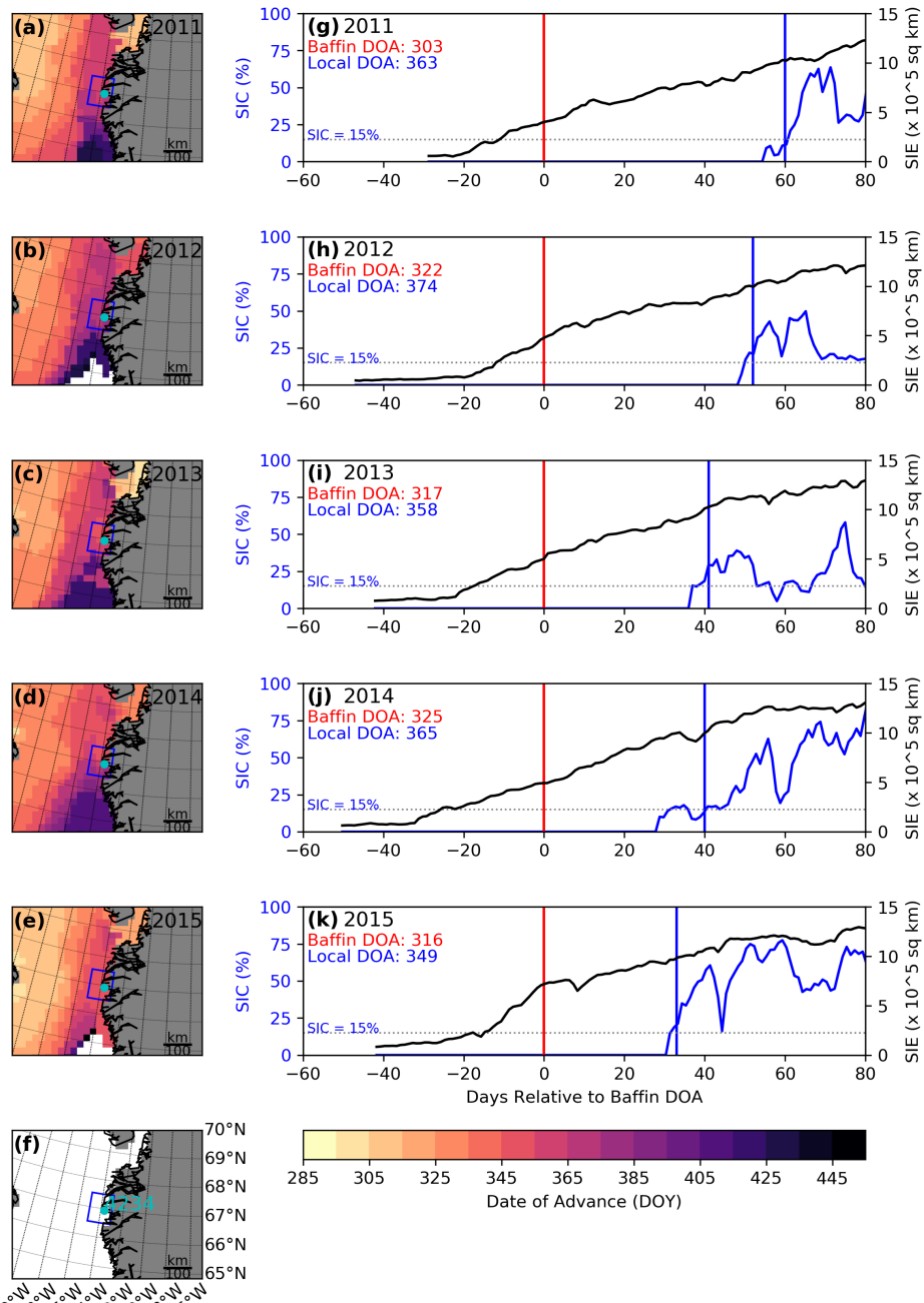

**Figure 3.** Maps of the Baffin Bay date of sea ice advance (DOA) for a-e) 2011-2015. The location of Sisimiut (WMO code 4234) is marked in cyan, white indicates locations where DOA was not observed and grey indicates land. Panel f) shows reference grid coordinates for maps a-e. DOY after 365 (or 366 in 2012) indicates that DOA occurs after 1 January the following year. Time series of sea ice extent (SIE) for the Baffin Bay region (black) and mean sea ice concentration (SIC) for the local domain (blue), 66.5 to 67.5°N and 53 to 55°W (blue polygon shown in a-f), relative to the Baffin-wide DOA (vertical red line) for g-k) 2011-2015. The vertical blue line shows the mean DOA for the local domain. The horizontal dotted line represents the 15% SIC threshold used to identify the DOA.

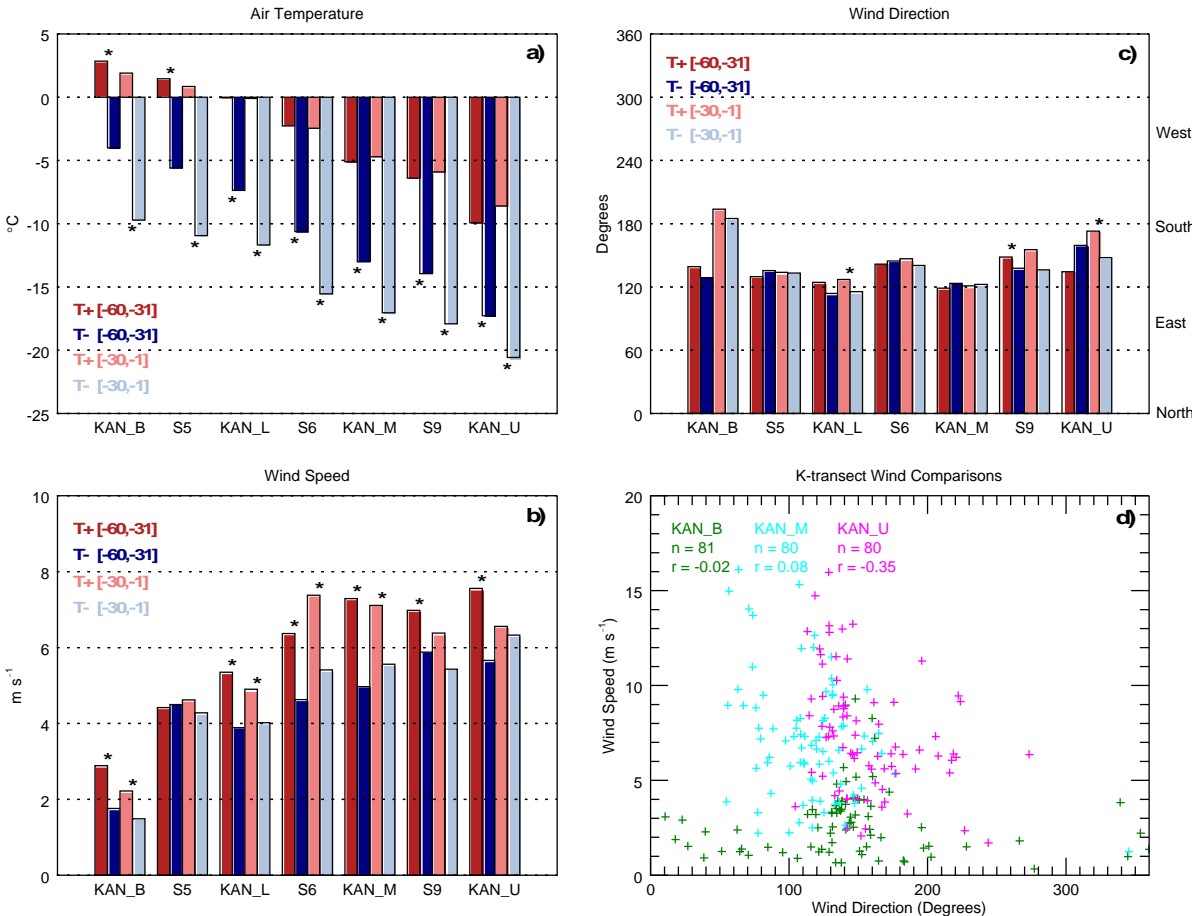

**Figure 4.** Composites of a) near-surface air temperature, b) wind speed, and c) wind direction for the T+ and T- events at KAN_B preceding the Baffin Bay date of sea ice advance (DOA), 2011-2015. Significant differences (p≤0.05) between T+ and T- composites over similar time windows are shown by an asterisk (*) between the bars. Panel d) shows daily mean wind speed as a function of direction for select, roughly equidistant K-transect PROMICE stations.

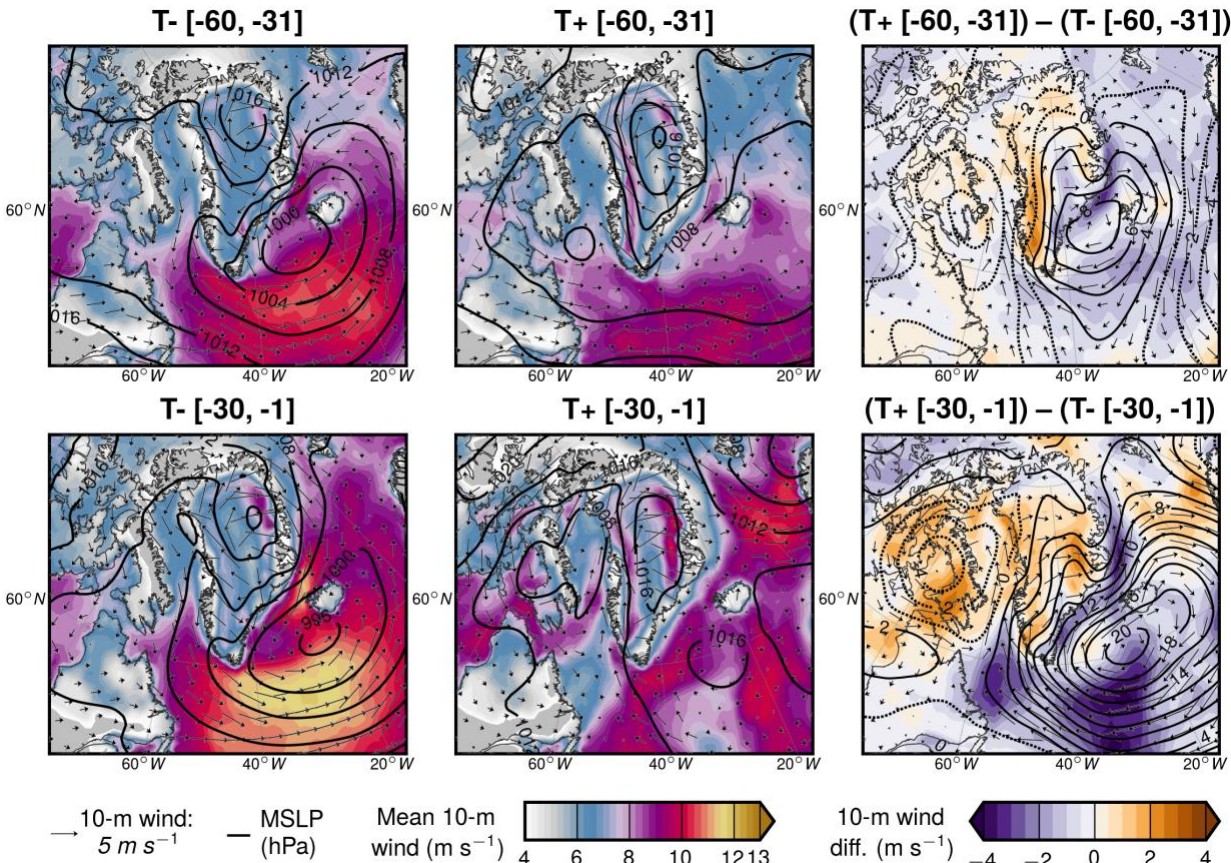

**Figure 5.** North Atlantic mean sea-level pressure (MSLP) and 10-m wind composites from ERA-Interim for T+ and T- events at KAN_B and their differences for the two periods preceding the Baffin Bay date of sea ice advance (DOA).

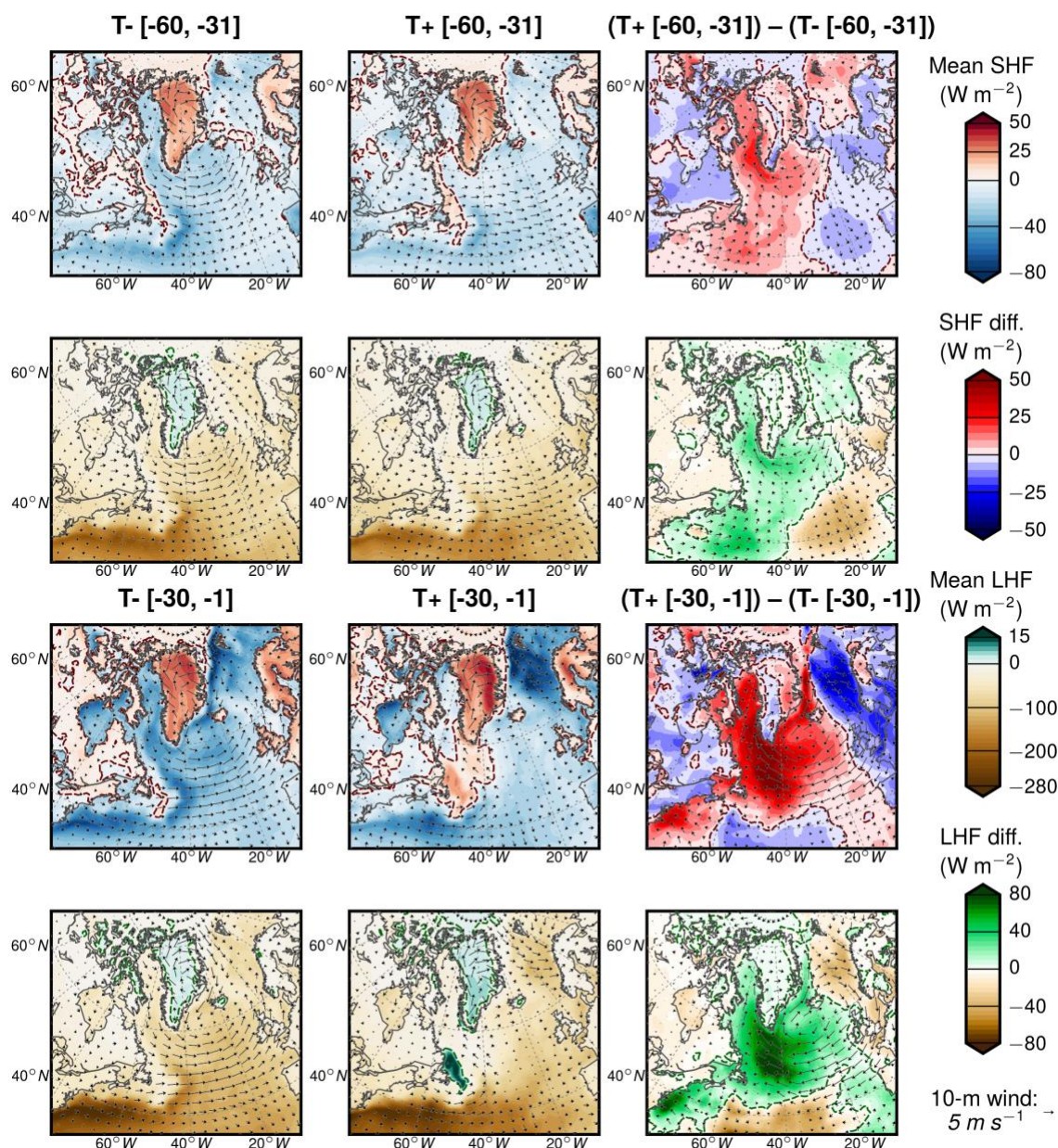

**Figure 6.** North Atlantic sensible heat flux (SHF), latent heat flux (LHF), and 10-m wind composites from ERA-
Interim for T+ and T- events at KAN_B and their differences for the two periods preceding the Baffin Bay date of sea
ice advance (DOA).

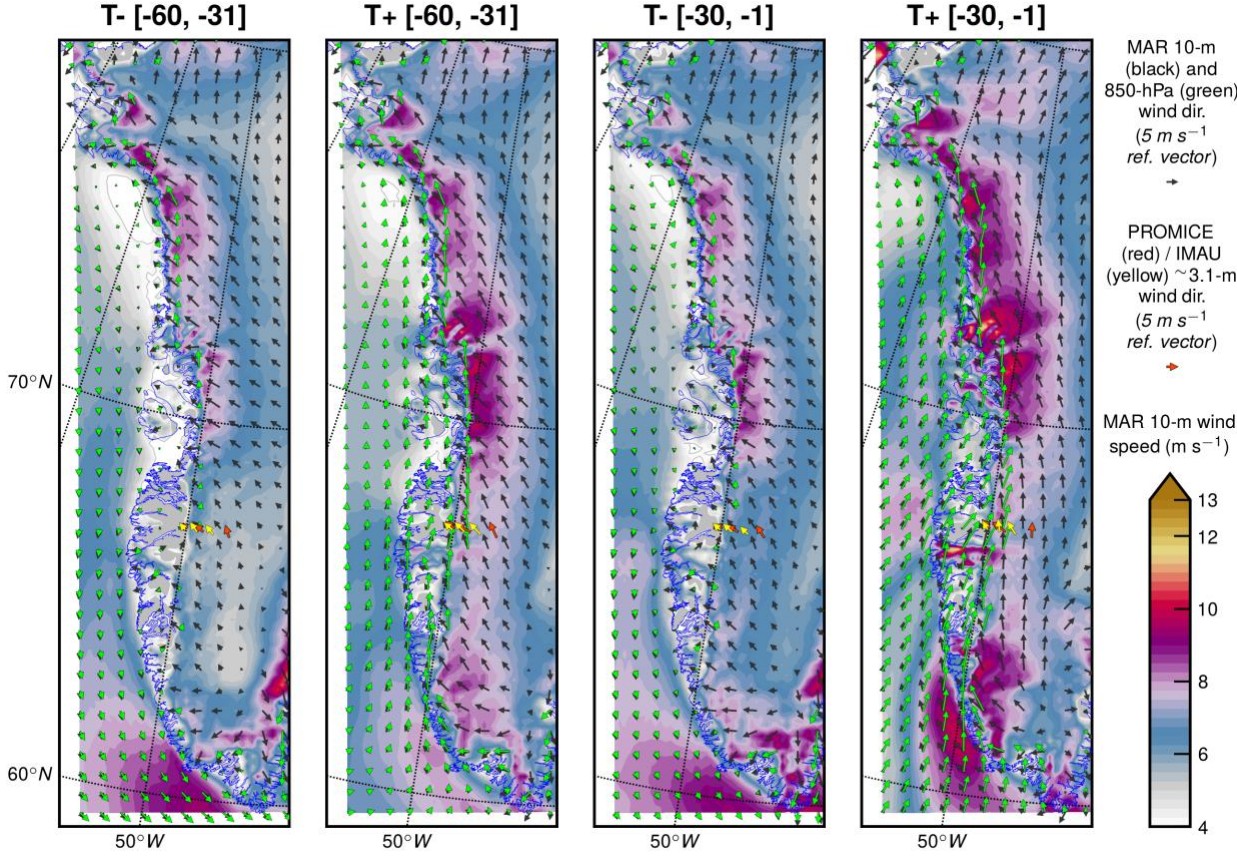

630

**Figure 7.** Composites of MAR 10-m (black arrows) and 850 hPa (green arrows) vector winds for the T+ and T- events at KAN_B preceding the Baffin Bay date of sea ice advance (DOA), 2011-2015. Wind observations from PROMICE (red) and IMAU (yellow arrows) are overlaid for reference.

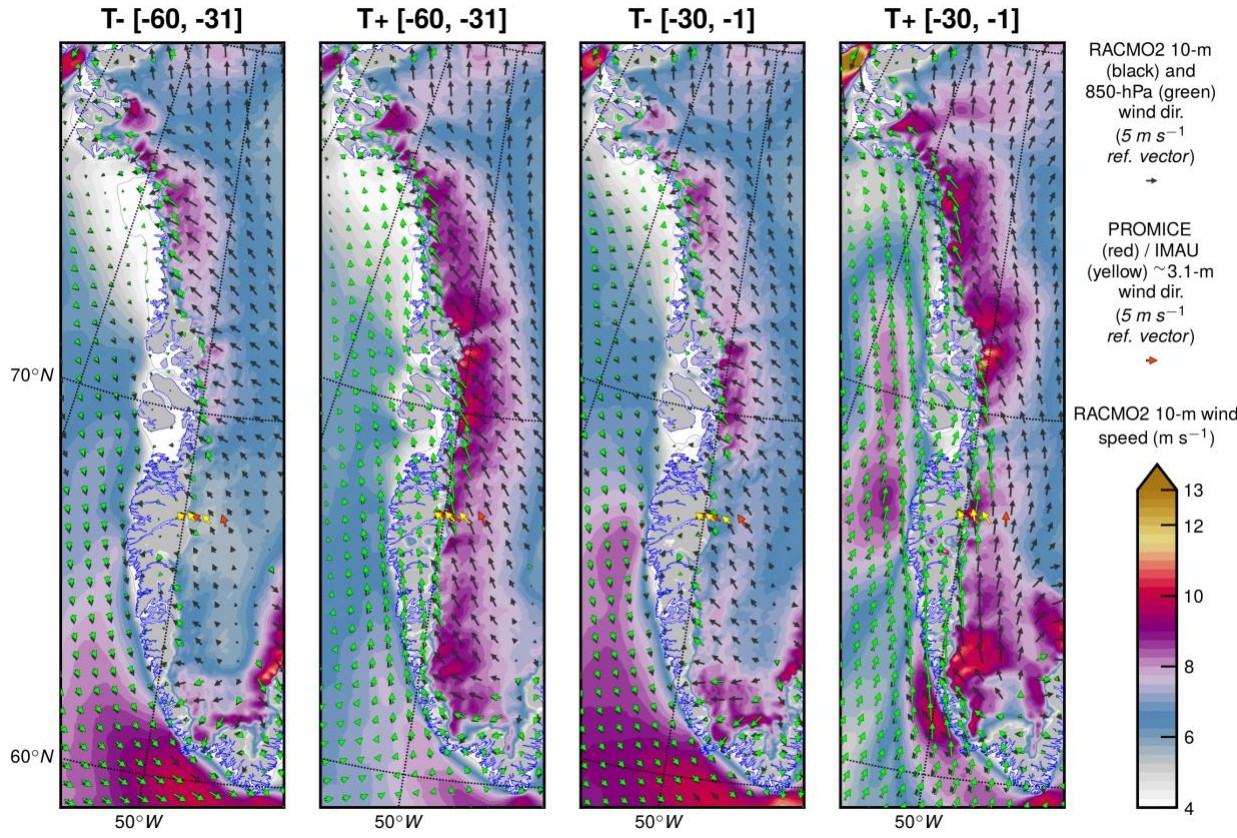

**Figure 8.** Composites of RACMO2 10-m (black arrows) and 850 hPa (green arrows) vector winds for the T+ and T-events at KAN_B preceding the Baffin Bay date of sea ice advance (DOA), 2011-2015 (refer to methods for details). Wind observations from PROMICE (red arrows) and IMAU (yellow arrows) are overlaid for reference.

635

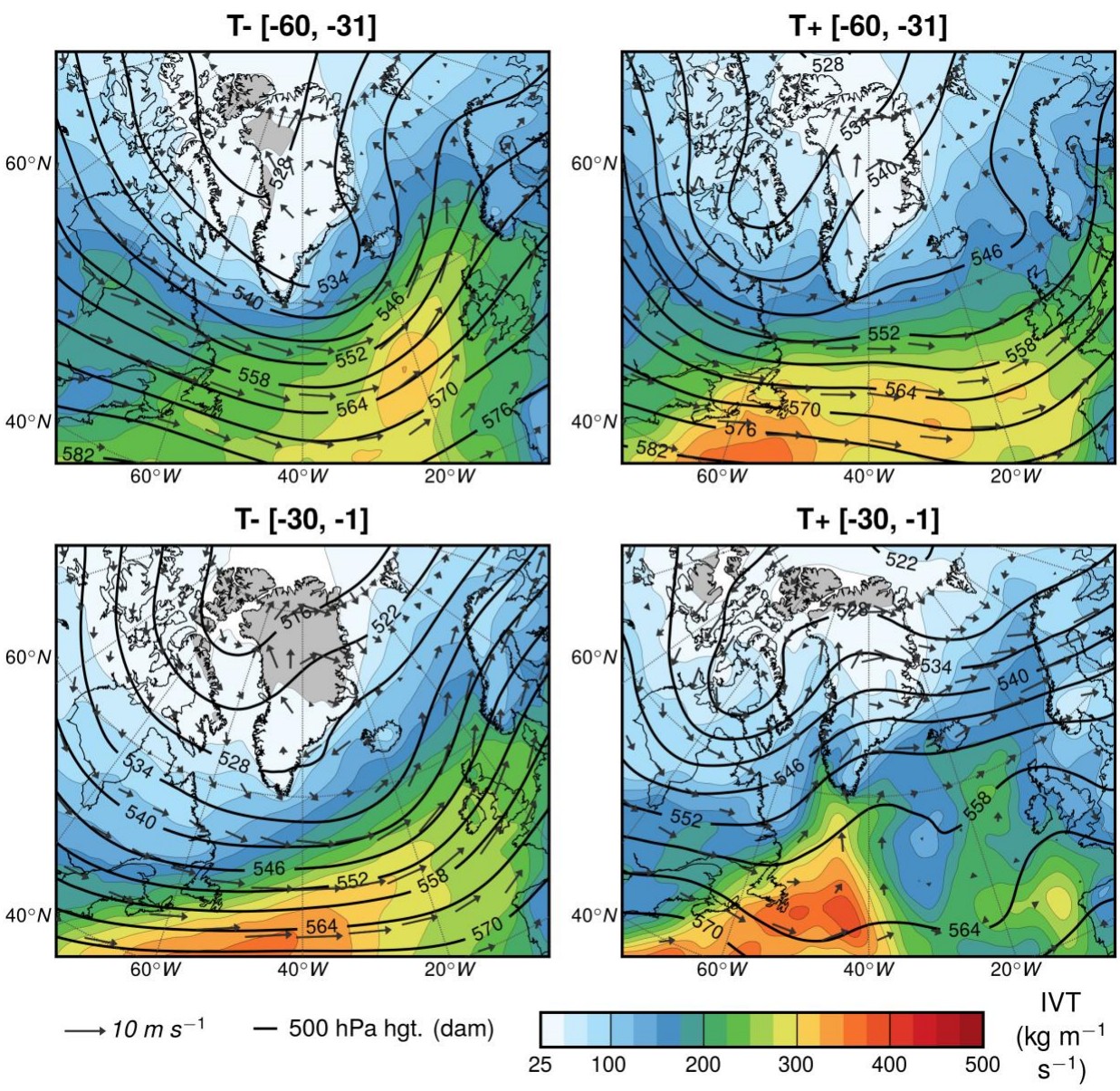

**Figure 9.** Composite plots of integrated vapor transport (IVT), 1000-700 hPa winds, and 500 hPa GPH from ERA-Interim for T+ and T- events at KAN_B for the two periods preceding the Baffin Bay date of sea ice advance (DOA).

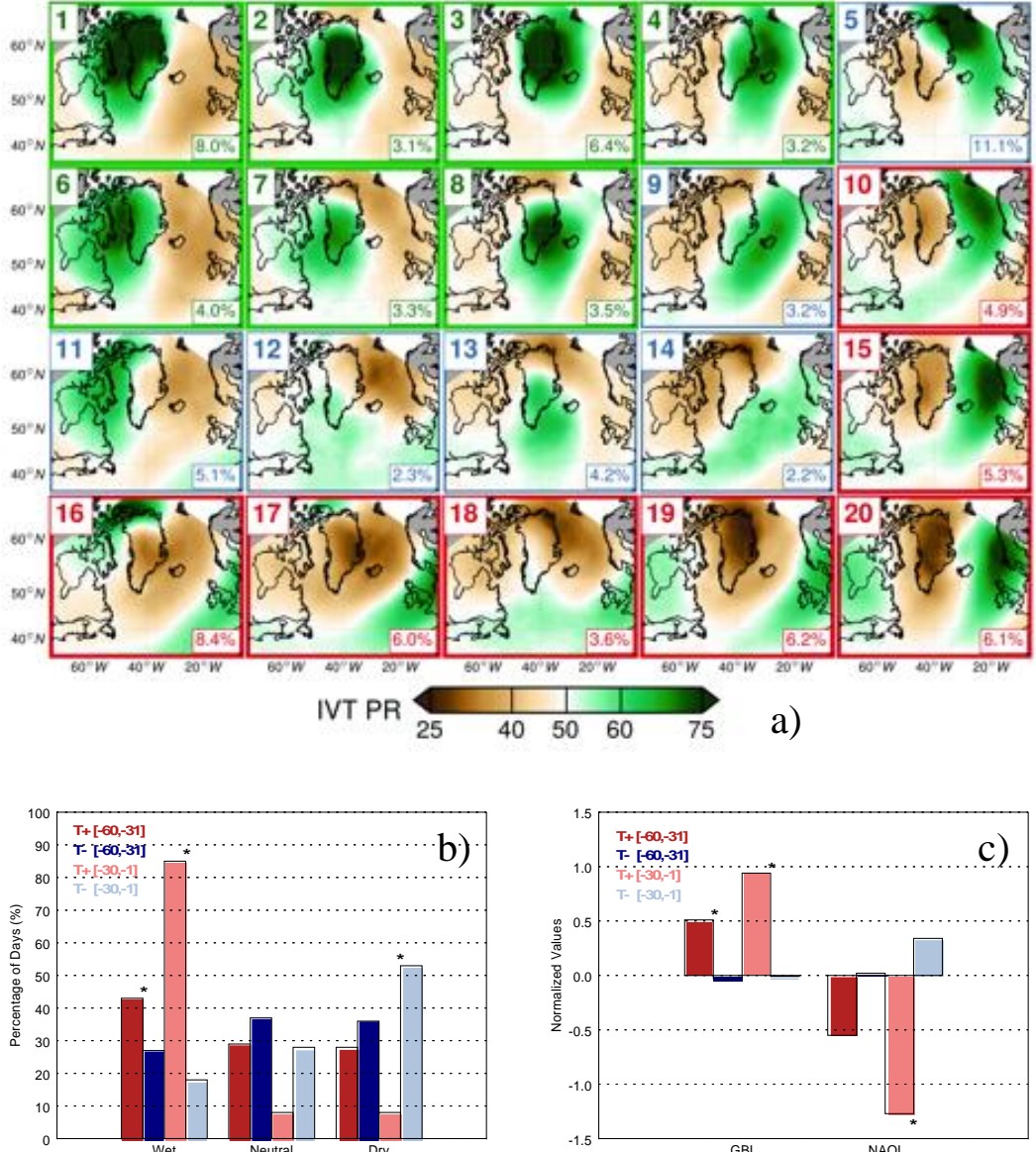

**Figure 10.** Percentile rank of integrated vapor transport fields (IVT PR) from ERA-Interim classified using a self-organizing map (SOM) approach a) and composites of b) IVT PR SOM node frequencies by wet, neutral and dry types (%). SOM aggregates in panel b) represent the ratio of each pattern's occurrence to the sum of all patterns for each time period and similarly colored bars sum to 100%. Composites presented in panel c) represent normalized Greenland Blocking Index (GBI) and North Atlantic Oscillation (NAO) values (unitless) for T+ and T- events at KAN_B for the two periods preceding the Baffin Bay date of sea ice advance (DOA). Significant differences (p≤0.05) between T+ and T- composites over similar time windows are shown by an asterisk (*) between the bars.