# Peer review of "Greenland Ice Sheet late-season melt: Investigating multi-scale drivers of K-transect events"

_The Cryosphere, 2018_

## Referee Comment (RC1) · Anonymous Referee #1 · 6 Feb 2019

—— General Comments ——

In this study, Ballinger et al. investigate potential mesoscale and synoptic drivers of late season melt events in the K-transect sector of southwest Greenland. They find that synoptic scale processes are the primary driver, with late season K-transect melt events characterized by southerly flows of warm, moist air from lower latitudes onto the ice sheet. In contrast, local ocean-atmosphere interactions over Baffin Bay are not found to have any direct role in driving late season melt events.

This study provides novel and valuable insights into the variability of GrIS melt, identifying an important link between synoptic scale moisture transport and unseasonal melt events, which has not been previously explored. Overall, the analysis is well done and the manuscript is well written. My main concerns are: (i) the robustness of the results

given the small number of years analyzed, (ii) lack of clarity regarding the initial hypothesis of marine layer forcing (what exactly would be expected if melt events were driven by marine layer forcing, and how do the actual results differ?), and (iii) potential issues with the interpretation of turbulent heat fluxes and associated arguments about the origin of warm, moist air that is transported from the south onto the GrIS. I discuss each of these below, along with other minor comments and suggestions for improvement.

––– Specific Comments –––

L32 "For the unseasonal melt period preceding the DOA": Which unseasonal melt period? This seems to imply a sustained period of unseasonal melt before the DOA, whereas the unseasonal melt usually occurs in transient events. I suggest rephrasing as simply "For the period preceding the DOA".

L40 "While thermal conduction and advection off south Baffin Bay open waters impact coastal air temperatures, consistent with previous studies": Where is this shown in the manuscript?

L47 "The Greenland Ice Sheet (GrIS) surface mass balance (SMB) decrease has contributed roughly 0.5 mm year-1": It's the overall mass balance decrease (SMB + ice discharge) that has contributed 0.5 mm/year, not the SMB.

L50 "west Greenland waters" -> "ocean waters west of Greenland"?

L83 "recent, temporally-anomalous GrIS late melt events spanning the end of summer to freeze-up of adjacent ocean waters": Which recent events? Please provide a few examples and references.

L112 and L156: Can you provide some further explanation as to why KAN_B was selected to build T+/T- composites, thus limiting the study period to only 5 years? Especially since the objective is to study melt events, why choose a station on the tundra instead of a station on the glacier? KAN_B has a much shorter data record than all the other K-transect stations, so the selected study period 2011-2015 is only a small sub-

set of the available data. With the short study period and with composites sub-divided into D_60 and D_30 bins, the D_30 composites comprise only 13 days of data.

The next closest station, S5, has data going back to 2003, which would allow for a more robust analysis over a 13 year study period. If you repeat the analysis using S5 to build composites over 2003-2015, do the results shown in Figs. 3-6 and S4-S5 change very much? I think it would be preferable to use station S5 and the longer study period for these figures, or at least mention whether the KAN_B-based results are robust when S5 and the longer period are used. The only place I see a need for a shorter study period is in Table 2 and Fig. 2, where all the K-transect stations are compared over an overlapping period, but if KAN_B were excluded then I think the overlapping period could still extend back to 2008 instead of 2011.

L163: It would be helpful to summarize from Table 2 the frequency of T+ events at KAN_B (or S5, if it were used instead of KAN_B to build composites), and mention what percentage of days each represents, i.e., the D_60 bin has 69 events ($\sim$46% of days) and the D_30 bin has 13 events ($\sim$9% of days).

L176 "IVT PR is then classified using the SOM technique to produce a matrix of moisture transport patterns, or nodes, that typically occur over the Greenland region": This is very vague. Some additional description should be included for the benefit of readers who are not familiar with SOM - for example, mention that it is an unsupervised machine learning algorithm and that each daily IVT PR field is classified with its closest matching node from the SOM.

Are the SOM dimensions and training parameters the same as in Mattingly et al. (2016), or is anything different? It's not clear from this description. It would also be helpful to include in the Supplemental Materials a figure showing the SOM nodes and the wet/dry/neutral groups, so that readers have a better understanding of the SOM classification used in this study.

L178 "As in Mattingly et al. (2016), similar wet (anomalously high), neutral (near climatological median values), and dry (anomalously low) IVT patterns are aggregated, then their frequencies are composited and tested following methods previously summarized": What is meant by "tested" in this context? In the cited paper, the clustering of nodes into wet/dry/neutral groups was subjective, based on visual inspection. I don't recall any testing of this grouping procedure.

L208-231 I find it difficult to follow the reasoning in these two paragraphs that leads to the conclusion that "late season melt inferred from T+ events may be driven by synoptic patterns as opposed to local marine forcing" (L234). Why exactly do we reject the hypothesis that local marine forcing drives the T+ events? Is it simply because the K-transect wind directions in T+ events are offshore instead of onshore? What about the other aspects discussed here – such as K-transect wind speeds, sensible and latent heat fluxes over ocean and GrIS, near surface winds in eastern Baffin Bay, etc. – how do they contribute to this argument?

These paragraphs should be revised to clarify: if local marine forcing were driving the T+ events, then what would we expect the results here to look like? How do the actual results differ from that?

L214-218 and Figures S4-S5: Sensible and latent heat fluxes in Figures S4 and S5 are described here as positive in the upward direction. Is this correct? I would have expected sensible heat fluxes over GrIS to be directed from the atmosphere to the ice sheet surface (i.e. negative SHF if the upward direction is positive). In particular, in the K-transect region during T+ events, if air temperature is greater than 0 C but the melting ice surface is at most 0 C, how can SHF be directed upward?

L240 "In both T+ cases, low-level winds circulate poleward over north Labrador Sea areas of upward, turbulent heat flux (Figs. S4 and S5), aiding the heat and moisture transfer (as shown by heightened IVT values in T+ relative to T- ) over western Greenland during D60 and D30 (Fig. 5)": Following from my previous comment about the direction of SHF and LHF in Figures S4 and S5, I am wondering if turbulent fluxes over

the north Labrador Sea are actually upward and if this region is actually the source of the heat and moisture transported onto western GrIS? Or do the heat and moisture originate from further south, around 30-35S? Subtropical origin seems more likely and is more consistent with the IVT fields shown in Figure 5.

L254 "higher 500 GPH values and on-ice lower tropospheric mean winds": The phrasing here seems confusing. "Higher" refers to GPH being higher in T+ relative to T-events, but does "lower" refer to wind speeds in T+ relative to T-, or does it mean "lower troposphere"?

L267 and Figure S6: Since each station-year is a separate data point, why not include all years of data for each station in Fig. S6 and related discussion? Only using the 2011-2015 period gives a very limited perspective on whether the temperatures are typically correlated or not, since we see statistically significant correlations in only three of five years.

—— Typographic Corrections and Figure/Table Formatting ——

Headers for sub-sections 2.2-2.4 should be in bold for consistency with the other headers.

Table 2: I find the labels here a bit confusing. Although each row is labelled as a relative measure (e.g., "S5 vs. KAN_B"), the columns "T+ n[-60, -31]" and "T+ n[-30, -1]" are not relative to KAN_B, they are the actual counts for each individual station.

Figures 2 and 6: It's difficult to distinguish between the dark and light red/blue colors in the bar charts.

Figures 3-4: It's difficult to distinguish cyan arrows over blue background. It's also difficult to see the wind vectors for the K-transect stations - perhaps these arrows could use heavier line weights and/or different colors?

Figure 5: Many of the GPH contour labels are cut off at the edges of the figures.

L518: The last four references are not in alphabetic order with the rest of the bibliography.

---

## Referee Comment (RC2) · Charalampos Charalampidis (Referee) · 11 Feb 2019

General comments. The Discussion Paper by Ballinger et al. is an interesting and novel attempt based on in situ, Automatic Weather Station (AWS) observations and Regional-Climate Model (RCM) output to identify the prevailing atmospheric-circulation pattern that causes unseasonal melt events in the southwestern Greenland ice sheet, and whether sea-ice formation and related turbulent-heat production over Baffin Bay in early autumn is of importance.

The study is characterized by an accurate and well-structured methodology, but in my opinion, not also interpretations. As an example, the authors conclude that they "find no evidence to support the hypothesis that local open water and resultant turbulent heating

has a demonstrable impact on inland ice melt events", even though the approach is purely statistical; the study does not disprove the hypothesis, but rather proves that it is not the dominant mechanism.

Structurally, the study suffers from obvious overlooks and requires attention. As examples, I note the allegedly examined AWS pressure and humidity according to the Abstract, even though the analysis is based on AWS air temperature and wind properties only, and the mention of a Sisimiut AWS in the Results and of an Upernavik AWS in the Discussion, without being previously introduced or presenting any related figure in the main body.

I recommend a major-revisions status, primarily to provide the authors with ample time to improve the structure and reasoning throughout the text. I also have a comment on the methodology concerning the AWS analysis that, in my opinion, needs to be addressed before publication.

Specific comments

The linguistic level is good, however sentences tend to be condensed, and would benefit by a more elaborate style. Since the study is not purely meteorological and aims potentially to appeal to a wider Earth-sciences audience, some effort should be invested into familiarizing readers with meteorological norms instead of just mentioning parameters, numbers, and units. Additionally, the reader's knowledge over methodological approaches is taken for granted. As a rule of thumb, a second-year Earth-sciences master's student should be able to follow without referring to textbooks for clarifications.

The study has a broadly clear structure, but the argumentation is not straightforward, and complicated – sometimes even speculative – discussion points are widespread in Results and Conclusions. Also, as a rule of thumb, referencing supplementary material in Results or other studies in Conclusions should be avoided, in order for the study to be self-standing.

[Figure]

Effort should be invested into describing the theoretical background and the study area, particularly its wind regime and mechanisms, as well as recent Surface Mass-Budget changes within the Introduction or before Data as a self-standing section, so that readers can easily relate afterwards to the Discussion. Recommended literature: Van Angelen et al. (2011), Van As et al. (2014), and the classic GIMEX-90/91 work. The inclusion of a description of the different areas of a glacier, as well as an Equilibrium-Line Altitude definition and how high it is in the study area, is also recommended.

The use of KAN_B observations as basis for the analysis needs to be motivated better than just saying that it records more above-zero Celsius air temperatures than all the other K-Transect AWSs. The way I see it, by the end of August, after the end of the melt season, the increased solar-zenith angle and ice-sheet surface albedo limit the influence of solar radiation on potential ice-sheet surface melt, which is thereafter driven by climatic drivers influencing downward longwave radiation (i.e. cloudiness) and turbulent fluxes (e.g. September 2010; Charalampidis et al., 2015). Also, katabatic-wind flow, driven by the balance of downward sensible heat flux and radiative cooling at the surface, becomes gradually colder and denser, and therefore more intense and laminar, thereby hindering turbulent mixing over large part of the AWS transect that is characterized by decreased surface roughness (Smeets and van den Broeke, 2008; see also specific comments for L208-209 and L211-218). Hence, melt events might occur in cases of weak katabatic-wind flow and concurrent northwesterly atmospheric advection from Baffin Bay, or moderate to strong katabatic-wind flow and concurrent southerly barrier winds along the ice-sheet margin driven by synoptic circulation.

Located only 1 km away from the ice-sheet margin, KAN_B should be the station most sensitive to both these circulation patterns. In both cases, the local boundary layer will become more humid (the role of humidity on energy balance and as a cause of surface melt should be explicitly described), and KAN_B might record positive air temperatures, in which case it is also worth inspecting all on-ice AWSs for positive air temperatures, which would be indicative of surface melt. (It should be explicitly mentioned in the

text that positive near-surface air temperature is used as an indicator of melt at the ice-sheet surface.)

The statistical analysis should reveal which of the aforementioned cases is the most frequent (and not simply differentiate between regional meteorological processes acting during positive and negative KAN_B air-temperature events, as currently stated in L154-155. What goes on during negative KAN_B – and hence K-transect – air temperatures is not the focus per se; it is just used as means of comparison). The above theoretical introduction material along with Figure 2 alone proves, in my opinion, that synoptic circulation in the South of Greenland is the driver of unseasonal melt events mostly in the lower ablation area until the elevation of KAN_L. The persistent katabatic-wind regime acts otherwise as a shield against turbulent forcing from Baffin Bay. All the rest should be an elaboration around this key result.

It might be insightful to make a similar comparison of KAN_B with observations further away from the ice-sheet margin. I do not know what the result might be, but I am guessing weakening southerly wind away from the margin in case of synoptic-driven, barrier-wind occurrences, and otherwise more intense winds than whatever KAN_B is reporting at that time, directed toward the ice sheet.

Nevertheless – and this is my major methodological concern – it is not clear in the text how the KAN_B temperature events are defined. Are these definitions based on hourly or daily observations? Is there a time window plus/minus delta_t around a warm/cold KAN_B event t within which the conditions at all other stations are evaluated? Is delta_t selected larger or smaller depending on the duration of the KAN_B event t? Eventually, how sensitive are the statistics, and how conclusive the implications, between hourly and daily analysis? Please, elaborate.

Technical comments

Please, make use of dashes, or hyphens if appropriate, throughout the text to facilitate readability in case of several nouns in a sequence, e.g. in the Abstract alone

L27: air-temperature episodes; L29: open-water duration; L30: sea-ice advance; L31: sea-ice growth; L38: late-season, ablation-zone melt events, but also L129: "Surface*hyphen*atmosphere features" Also, differentiate between dashes and minuses.

Use the more appropriate "area" instead of "zone" throughout the text.

I recommend the use of "observations" instead of "data", while differentiating between in situ and remotely sensed, and "simulations" for modeling products.

I recommend the use of "average" instead of "mean".

Except for "Buffin DOA", complete "Buffin" as "Buffin Bay" throughout the text.

Capitalize the first letters of every word when an abbreviation is introduced, so that the reader's eye can make a quick connection, i.e. L30: Date Of sea-ice Advance (DOA).

L27: delete "significant", and reserve it exclusively for describing correlations.

L32: "unseasonal melt events. . ."

L34: "Southwest" or "southwestern"

L34: The influence of synoptic and mesoscale systems on the above- and below-freezing near-surface air-temperature events. . .

L35: "AWS" -> "the"; "The in situ observations. . ."; Why are pressure and humidity observations mentioned here? I was unable to spot them later in the study. Why is wind speed and direction not mentioned?

L36: "against" -> "with"

L35-38: I suppose "MAR, RACMO2, and ERA-Interim are used to provide context to the in situ observations by explaining the air-mass origins and the (thermo) dynamic drivers of melt events." As it is written now, it gives the impression that the in situ observations need to be calibrated against RCMs.

L41: Delete "consistent with previous studies". Try not to refer in the Abstract and

Conclusions to other studies.

L42: Consider removing "pressure-gradient driven" from the Abstract, and describe briefly the mechanism in the Introduction.

L42: "Katabatic-wind regime", since the word "katabatic" is an adjective referring to a descending object, in this case, wind.

L41-42: I think this is an overall complicated explanation. I think the primary obstacle of Baffine-Bay marine-air intrusions over the ice sheet should be intuitively the persistent katabatic wind regime that intensifies in the beginning of autumn. Additionally, barrier winds might be present, in which case melt might occur due to air-mass transport from the Atlantic. I propose restructuring as: "…are obstructed by the persistent katabatic-wind regime flowing downslope from the ice sheet, and the occasional occurrence of barrier winds along the ice-sheet margin.", or something along these lines.

L44: "Substantial mass losses…"

L46: "have become sensitive" or "are becoming increasingly sensitive"

L48: Since all Van or Van de Author have been categorized under V in References, consider capitalizing the first letter of "Van" in the in-text occurrences, as well as that of the first author in the References, in order to allow the reader to navigate through easily.

L53: Replace "play a key role" by "are of importance"

L54: "surface melt at the southwestern GrIS."

L55: "Conflicting evidence have been presented in literature over the past decade regarding…"

L56: "Regional-climate simulations…"

L60: "noted that…"

L61: Replace "which" by "and"; replace "by" by "to".

L63: "in Ilulissat and Nuuk, approximately 200 [Unit] to the north and south of Kanger-lussuaq, respectively. . ."

L66: Replace "coupled" with "correlated".

L67: Please, delete "robust". Statistical quantities are a matter of context, and their interpretation is dependent on the datasets and the research question itself, which in this instance are not present; "(from Markus et al., 2009)"

L69: "The authors found that significant, positive correlations between Baffin and Labrador SST and coastal SAT often persist. . ."

L70: "onset of freeze" should suffice; "Applying a similar correlative approach on. . ."

L71: "while utilizing melt/freeze product. . ."

L74: "Both studies indicated. . ."; replace "upper-level" with "high-altitude".

L75: Replace "at the limits of" with "toward the end of".

L81: Replace "research studies" with "literature".

L86: Delete "automatic weather stations", since AWS has already been introduced.

L99: "Daily observations are available over the 1979–2015 period at a. . ."

L101: Delete "day of sea-ice advance", since DOA has already been introduced.

L102: Bliss et al. (in review)

L112: "in this study (Fig. 1)."

L113: "KAN_B that is situated approximately 1 km away from the ice-sheet margin. . ."

L117: Delete "tundra (KAN_B only) or glacier ice"

L116-117: The observational distance from the surface decreases during winter season due to the accumulating snowcover around the AWSs by as much as two meters depending on the location along the transect, while it increases again each melt season until the complete ablation of the accumulated snowpack. In the case of KAN_U that is located in the lower accumulation area, this distance tends to become shorter over the course of a few positive mass-budget years due to the incomplete ablation of the snow cover, but also due to the sinking of the AWS tripod during ablation in the temperate firn below, as discussed by Charalampidis et al. (2015). After the extreme 2012 melt season, snow accumulated around KAN_U during the 2013 and 2014 positive mass-budget years, while on 3 May 2015 the half-buried tripod was replaced.

L121: Consider including also Citterio et al. (2015) describing the PROMICE AWS.

L125: Delete "have been shown to"

L127: "from between 1000–200 hPa height, corresponding to the distance between sea level and lower stratosphere at 67 N (e.g. Zängl and Hoinka, 2001)..."

L129: "Surface*hyphen*atmosphere" since the two are related/interacting.

L130: Replace "features" with "interactions". Alternatively, replace "Surface–atmosphere interactions" with "Boundary-layer processes"; 500 hPa height (i.e. ∼5000 m above sea level)"

L132: "10-m above surface and 850 hPa height..."

L133: "low-level atmospheric flow..."

L143: "ice caps, and improving..."

L145: "to characterize near-surface..."

L151: It should be mentioned that above 0 C air temperatures at on-ice AWSs are considered an indicator of ice-sheet melt.

L152-165: It is not clear how the KAN_B temperature events are defined. I refer to

Specific comments.

L157: "due to the station's location. . ."

L171: "Based on RCM output, . . ."

L172: Delete "integrated water-vapor transport", since IVT has already been introduced.

L183: Refrain from mentioning supplementary material as part of the text, and only refer to them in parentheses at the end of sentences. Nevertheless, supplementary material should be referenced outside Results, and if Fig. S1 should be referenced in the first sentence of the Results, perhaps it belongs in the main article.

L189-191: Nice, but perhaps also mention that the difference has narrowed primarily due to the prolongation of the melt season over the course of the thirty years.

L193-195: It is a bit awkward to see all of a sudden a Sisimiut station being mentioned here in the first subsection in Results even though it has not been introduced earlier, and a case being made based on supplementary material. Please, restructure.

L195-199: This belongs in the Discussion.

L201: I am not sure I understand what this first sentence is trying to justify. Also, sounds like it belongs in Methods.

L203: Does the term "composite" essentially refer to the average of all instances? Please, clarify.

L206: Second half of the sentence refers to T above 0 C events? Please, clarify.

L208-209: South-southeasterly winds should have a 157.5 degrees direction, given the orientation of the PROMICE AWSs, so what is now written is incorrect. These winds in autumn are katabatic with a downslope direction (90 degrees) deflected 45 degrees to the right by Coriolis force (i.e. southeasterly direction; cf. Van den Broeke et al., 2009),

plus potential southerly synoptic influence. Nevertheless, this quantification seems to be more prominent above the long-term ELA and KAN_B. Good thing to also mention.

L211-218: This belongs in the Discussion. Nevertheless, I am not sure I agree with this interpretation, and I explain: Strengthening wind speeds during positive air-temperature events at KAN_B do not reflect strengthening of katabatic flow (i.e. cold, dense wind), but rather that additional wind components inducing turbulent mixing might be present. A strengthened katabatic flow would remain cold, and would be more laminar, and hence less turbulent, than usual. Katabatic wind could enhance turbulent mixing during melt season at the lowermost rough-surface parts of the transect, but we are discussing unseasonal melt events, implying that sunlight is reduced, hence surface melt is not sustained in the way it does during the melt season, while surface roughness below ELA might be substantially decreased due to accumulating snow cover, and above ELA slightly increased for the same reason, i.e. sastrugi formation. The observations from most AWSs within the same averaging periods suggest southerly deviation during positive air-temperature events from the cold-event direction, and the way I perceive it, southerly synoptic influence. I note that Figure 2 and Table 2 suggest that melt might occur primarily at the two lowermost on-ice AWSs, while wind speed as well as wind-speed differences between positive and negative cases within the same averaging periods are more pronounced at higher elevations due to comparatively reduced surface roughness (cf. Van den Broeke et al., 2009).

L214-215: Positive energy flux that contributes to ice melt is most definitely downward, i.e. directed from the atmosphere toward the ice-sheet surface, and not necessarily the result of increased wind, rather the result of increased turbulence; "Sensible heat fluxes" does not sound nice. Consider removing the plural.

L219: The first sentence is not a result. Please, delete or relocate to Introduction.

L222: "at KAN_M and S9 in the upper ablation area."

L223: Replace "modeled" with "simulated".

[Figure]

L223-225: Difficult sentence, consider revising. Also, please, relocate in the Discussion.

L228-229: This sentence, as it is, belongs in the Discussion. You can reformulate and keep in the Results as: "We note that there is a height difference between the RCM 10-m above-surface output and the measuring heights of the AWSs, as mentioned in the Data section.", or something like this.

L233-234: This first sentence is Discussion material. Please, relocate or reformulate, since it sounds like a rather abstract introductory sentence referring to the previously outlined results.

L239: Explain the significance of 540 dam (i.e. 5400 geopotential meter), i.e. it often distinguishes solid and liquid precipitation.

L255: The link between moist air masses and how they may facilitate ablation-area melt is not clear. Please, elaborate preferably in the Introduction.

L261: Include a short comment on strong negative NAO phases, as seen in monthly observations.

L266: "turbulent atmospheric heating"

L267-269: Please, refrain from referring to supplementary material as main part of the text, i.e. delete "As shown in Fig.S6". Instead, include a citation at the end of the sentence. It should be mentioned that it is a moderate to weak link that seems to be year-dependent. The way I perceive it, this differentiation amongst years suggests synoptic-circulation control that may or may not be present each year, and that is a good comment to include before anything else. (For example, Fig. S6 suggests weak correlation in autumn 2013 between local sea-surface temperature and AWSs, when limited melt occurred even at KAN_U (September 2013; Charalampidis et al., 2015).)

L275: "statistically robust co-variability": Please, remove "robust" and reformulate.

L271-279: This part seems somewhat arbitrary, since no evidence of a different wind regime between 2011–2012 and 2013–2015 was presented. Also, I am not sure I agree with the categorization. Please, revise/clarify.

L287: Mention also the distance from Kangerlussuaq.

L290-294: This is highly unclear and speculative. Please, revise.

L295: "appears to be of minor importance. . ."

L304-306: Please, clarify.

L306: "Denmark Strait and Irminger Sea at the East coast of Greenland. . ."

L316: Generally, Conclusions outline key findings in the Results and key Discussion points. The current state appears more like just another part of the Discussion, while the only Concluding bits are between L319-324 and L341-345. Please, consider rewriting the whole section outlining important quantifications from Results. Please remove all citations, as the Conclusions should refer only to the present study, and should be self-standing.

L317-319: Consider simplifying; rephrase "around the limits"

L377: Delete "2018"

Table 1: Include a minus in front of the 1 km of KAN_B, so the reader can immediately see that it is different from the rest of the stations. "50*hyphen*150 m"

Table 2: Were these daily averages? Please, clarify. Note that dashes have been used instead of minuses. Also, make column lines between columns 3-4 and 5-6 thicker to facilitate the eye of the reader.

L540: Replace "30 to 60-day window" with "time window defined by the 60th and 30th day before day of ice advance (DOA)"

Figure 2: Include the legend that is shown in panel a also in b and c. In panel c, include also N-S-E-W. In panel d, are these points in reference to daily or hourly values? Please, specify.

L551: Define "composites" better.

L552-553: Difficult to fathom. Please, clarify.

L553: "selected"

L569: Spell out IVT and include abbreviation since it is used as such in the figure.

Figure 6: Include legend in both panels. In the caption, spell out all abbreviations and include abbreviation in parentheses.

L754: Define "composites" better.

L577-578: Difficult to fathom. Please, clarify.

Table S1: Dashes as minuses. Please, correct. "2011*hyphen*2015"

Figure S2, S3, S4, S5: Spell out DOA initially, and then use abbreviation.

Figure S4, S5: Define composites better.

Figure S6: Introduce SST properly in caption.

References

Charalampidis, C., van As, D., Box, J. E., van den Broeke, M. R., Colgan, W. T., Doyle, S. H., Hubbard, A. L., MacFerrin, M., Machguth, H., and Smeets, C. J. P. P.: Changing surface–atmosphere energy exchange and refreezing capacity of the lower accumulation area, West Greenland, The Cryosphere, 9, 2163-2181, doi:10.5194/tc-9-2163-2015, 2015.

Citterio, M., van As, D., Ahlstrøm, A. P., Andersen, M. L., Andersen, S. B., Box, J. E., Charalampidis, C., Colgan, W. T., Fausto, R. S., Nielsen, S., and Veicherts, M.: Automatic weather stations for basic and applied glaciological research, Geol. Surv.

Denmark Greenland Bull., 33, 69–72, 2015.

Van Angelen, J. H., van den Broeke, M. R., and van de Berg, W. J.: Momentum budget of the atmospheric boundary layer over the Greenland ice sheet and its surrounding seas, J. Geophys. Res.- Atmos., 116, D10101, doi:10.1029/2010JD015485, 2011.

Van As, D., Fausto, R. S., Steffen, K., and the PROMICE project team: Katabatic winds and piteraq storms: observations from the Greenland ice sheet, Geol. Surv. Denmark Greenland Bull., 31, 83–86, 2014.

Zängl, G., and Hoinka, K. P.: The Tropopause in the Polar Regions, J. Climate, 14, 3117–3139, doi: 10.1175/1520-0442(2001)014<3117:TTITPR>2.0.CO;2, 2001.

---

## Referee Comment (RC3) · Anonymous Referee #3 · 17 Feb 2019

General comments:

Ballinger, et al propose a hypothesis for the onset of late-season melt events on the Western Greenland ice sheet. Some prior literature has suggested that open-water ice-free conditions in Baffin Bay cause melt incursions onto the ice sheet, but Ballinger, et al use weather stations data, regional climate models and reanalysis products to make a case that incursions of North Atlantic warm air are responsible for these late-season events, regardless of sea-ice conditions.

Overall, the hypothesis is novel, and I applaud the authors for bringing in multiple datasets to support it. As one of the other reviewers also noted, I am, however, very concerned that the short time series (2011-2015) limits the conclusions that can be made with any statistical certainty. From the evidence given, I would say the study

is suggestive, but not conclusive in its arguments. In my opinion, the authors have a choice: either use a longer time-series of measurements or model data (for instance, keep the 2011-15 analysis of AWS data but make the regional climate data analysis span from 1979-2015, which is possible given that the two datasets aren't even used in direct comparison to each other), *or* soften the conclusions to make it clear they are suggestive but not entirely conclusive given the short time series. With only 5 years at your disposal, random noise can easily be interpreted as interesting new patterns.

As always, I acknowledge that the authors have spent more time considering this paper than reviewers spend reading it. If authors believe my judgements are unfounded or based on a misunderstanding, or if I just missed something in a comment, they may of course make that case upon resubmission to the reviewers or editor. Overall I do think the study, and the hypothesis put forward, is quite suggestive and worth publishing for that reason, even with the concerns cited, but the concerns should be addressed.

Specific comments:

L31-32: "For the unseasonal melt period preceding the DOA" . . . this make it sounds as if it's referring to a *specific* melt event that hasn't yet been mentioned. Generalizing the sentence more to something like "For periods of unseasonal melt preceding the DOA,. . ." may be clearer to the readar.

L34-35: "the above and below freezing surface air temperature events. . ." same comment as above.

L83-84: "temporally-anomalous GrIS late melt events" : in much of the text, terms such as "unseasonal" and "temporally-anomalous" are used, but never really defined. Does is refer to melt events after a certain date/season? Recent late-season melt events that are more frequent than previous periods? The descriptor is a bit vague here.

L104: (quick format check) Several headings are not "bolded", while the rest are. Quick fixes.

L119: "erroneously low values are filtered out prior to analyses" : Which variables were filtered in this way: temperature, wind speed, others? If you can, be more specific.

L152: "A composite approach is applied to characterize atmospheric conditions…" : From what I saw, I couldn't find an explanation in the text of how these composites were put together, or what exactly the reader is looking at when seeing the composite numbers in the tables and figures. I don't expect the authors to provide a textbook lesson on composite statistics, but a brief 1-3 sentence explainer for the reader would help reproducibility and clarity of the text. Right now, if I took the same data and tried to reproduce the results I would have no guidance of how to perform the composite analysis other than the authors' word that they applied a "composite analysis." More specificity is needed here to make the results reproducible. If the method is identical to one used in other papers, a citation may suffice.

L162-165: The description of the timespans used in the composite approach is good, I would put this at the top of the paragraph before describing why composite approaches were used and what constraints were placed upon them.

L183: "Perspectives on… are shown" is a vague term, almost meaningless. If the sentence is meant to point out something specific that Figure S1 shows, describe it explicitly.

L183-199: The first several paragraphs of the Results refer to figures that are almost exclusively in the supplement (Figs S1-S3, with exception of the Fig. 1 map), which makes it awkward for the reader to follow along. These figures appear to be central to the results, not just supplementary. Unless you are limited by the number of figures, consider putting some of the these in the central text rather than forcing the reader to flip back and forth to an entirely separate document just to follow along with your argument.

L192-193: "Relative to the climatology… date in 2012-2015 (Table S1)." It is unclear why 2011 is separated from 2012-2015 here, or why this sentence exists at all. Interannual variability has always been present, and no evidence is provided that this juxta-position of years shows any more significant differences than any other random 5-year period in Greenland's climatological history (perhaps it is, but it isn't demonstrated). If not, just omit it.

L195-199: "Interannual differences… found in the east and north (Curry et al., 2014)." These sentences don't show new results, but discuss the context of other literature. Consider moving to Discussion.

L201-202: "The spatial coherence of observations across the K-transect along with inhomogeneous GrIS Region 3 spatial melt patterns and satellite pixel contamination issues at the tundra-ice interface, lead us to assess the melt events at the station level." It's very unclear what is meant by "inhomogeneous GrIS Region 3 spatial melt pat-terns", or why that would motivate a K-transect station-level approach. Please clarify. Similarly with "the spatial coherence of observations across the K-transect".

L208-209: "…and comparatively becomes slightly more southerly" One issue with the way these composite records are presented is that there are no uncertainties or spreads presented with them at all, other than which ones are/aren't statistically sig-nificant. Thus, comparisons between them, such as "slightly more southerly" are im-possible to make without knowing whether or not the difference are simply within the noise of the two datasets, or are just part of statistical noise. This is the case with all the comparisons, actually, and it makes the conclusions difficult to defend.

L223: "Modeled wind speeds are more intense during T+ versus T- events" How much more intense? 2 %? 150 %? As noted in the comment above, is the difference greater than statistical noise?

L227: "with low root mean squared errors (not shown)." Unless I'm misunderstanding this statement, RSMEs for two values should just be two numbers. They can be stated explicitly instead of just saying (not shown). If they weren't computed, this statement should be omitted.

L227: A slight positive bias in both models is evident. . ." I assume a slight positive bias in wind speed? Clarify so that the sentence can stand on its own without ambiguity.

L235: The "North Atlantic region" is not explicitly defined here, it is just vaguely referenced even though the authors are clearly looking at particular portions of the map. This ambiguity makes the rest of the entire paragraph extremely difficult to decipher from a reader's perspective, given that different portions of "the North Atlantic" behave in different manners in Figure 5.

L235-238: "Whereas T- events "left panels" tend to be characterized by northerly winds over the 1000-700 hPa layer, . . ." It is extremely unclear what is being referred to here. Are these northerly winds specifically in Baffin Bay, or over the whole figure, or. . . ?? Also, this sentence is disjointed and somewhat convoluted, its meaning unclear to the reader.

L238: "found over ice sheet" –> "found over the ice sheet"

L238-239: "with the 540 dam contour" The 540 dam contour is very unclear in most panels of figure 5. Consider making it clear to the reader exactly which contour is 540 without them needing to interpolate between other lines, if you're going to make a point specifically about the 540 line. Make it easy on your readers to see your point. (Also, as another reviewer noted, make contour labels fully visible, not partially cropped by the panel edge.)

L246-255: I found this paragraph particularly good and compelling.

L265: "toward earlier (later) melt (freeze)" I understand this sentence construct, and occasionally it's useful, but it's also extremely awkward to read and should only be used when necessary to save space. In this case, "toward earlier melt and later freeze" says the same thing with only one more word, and is much clearer English for the reader to understand.

L265: "This hotspot of melt" It's unclear if you're referring to a particular hotspot of melt

referred to earlier (if so, point it out), or making a more general statement here.

L267-269: "Sisimiut SSTs fluctuate... ablation zone at S9." It's unclear why this sentence is important. Additionally, it's unclear that a statistically significant difference can be inferred at all if it only happened in 3 of 5 years, but not the other two. Even with just 5 years of data it's hard to make firm conclusions about climate patterns... moreso if just picking three years selectively out of those five. This needs to be better justified, or omitted completely.

L271-279: The two points made at the end of this paragraph seem difficult to support conclusively with only 5 years at your disposal, comparing the first two years to the second three. The problem with such a short time series is that signals can be easily interpreted from random noise, making such conclusions problematic at best. This is emblematic of the greatest weakness of the whole paper, making strong conclusions about climate patterns from only five years of data. It is unclear that if you had a 30-year record (long enough to infer at least some of the variability of the patterns you describe), that the same inferences could be made in any significant way.

L289: "near-surface air penetrates at least to" –> "near-surface air often penetrates at least to" (There isn't any indication that it always does.)

L300-301: "positive (negative) GBI (NAO) values" –> "positive GBI and negative NAO values" (more readable)

L332-334: It is good that the authors recognize that a longer time series would help with these analyses. It is still unclear that all of the conclusions in the paper can be made so confidently with the short time series available, and that some of the results are more just "suggestive" than "conclusive."

─────────────────────

---

## Author Comment (AC1) · 11 Apr 2019

**Greenland Ice Sheet late-season melt: Investigating multi-scale drivers of K-transect events**

Thomas J. Ballinger, Thomas L. Mote, Kyle Mattingly, Angela C. Bliss, Edward Hanna, Dirk van As, Melissa Prieto, Saeideh Gharehchahi, Xavier Fettweis, Brice Noël, Paul C.J.P. Smeets, Carleen H. Reijmer, Mads H. Ribergaard, and John Cappelen

Manuscript submitted on 19 December 2018, revised on 10 April 2019

Author Response

We thank Charalampos Charalampidis and two anonymous reviewers for offering constructive and thorough comments on our paper. Each concern is addressed below and in the manuscript, and we believe these changes have improved the clarity and quality of the paper. In particular, we have taken care to address common concerns amongst reviewers that involve clarification of the composite methodology (please see Section 3 "Methods"). We note that co-author Carleen H. Reijmer was inadvertently omitted from the author listing in the discussion paper and has been added accordingly.

Reviewer comments are shown in *italics* while author responses are subsequently provided in red.

1. Anonymous Reviewer #1

*General Comments*

*In this study, Ballinger et al. investigate potential mesoscale and synoptic drivers of late season melt events in the K-transect sector of southwest Greenland. They find that synoptic scale processes are the primary driver, with late season K-transect melt events characterized by southerly flows of warm, moist air from lower latitudes onto the ice sheet. In contrast, local ocean-atmosphere interactions over Baffin Bay are not found to have any direct role in driving late season melt events.*

*This study provides novel and valuable insights into the variability of GrIS melt, identifying an important link between synoptic scale moisture transport and unseasonal melt events, which has not been previously explored. Overall, the analysis is well done and the manuscript is well written. My main concerns are: (i) the robustness of the results given the small number of years analyzed, (ii) lack of clarity regarding the initial hypothesis of marine layer forcing (what exactly would be expected if melt events were driven by marine layer forcing, and how do the actual results differ?), and (iii) potential issues with the interpretation of turbulent heat fluxes and associated arguments about the origin of warm, moist air that is transported from the south onto the GrIS. I discuss each of these below, along with other minor comments and suggestions for improvement.*

We thank the reviewer for their comments and have addressed their concerns in our revision.

*Specific Comments:*

*L32 "For the unseasonal melt period preceding the DOA": Which unseasonal melt period? This seems to imply a sustained period of unseasonal melt before the DOA, whereas the unseasonal melt usually occurs in transient events. I suggest rephrasing as simply "For the period preceding the DOA".*

Thank you for the clarifying suggestion; we have made this change.

*L40 "While thermal conduction and advection off south Baffin Bay open waters impact coastal air temperatures, consistent with previous studies": Where is this shown in the manuscript?*

Heat transfer off of ice-free ocean waters onto coastal areas is discussed at length in the second paragraph of the Introduction.

*L47 "The Greenland Ice Sheet (GrIS) surface mass balance (SMB) decrease has contributed roughly 0.5 mm year-1": It's the overall mass balance decrease (SMB + ice discharge) that has contributed 0.5 mm/year, not the SMB.*

We have made this clarification in the manuscript.

*L50 "west Greenland waters" -> "ocean waters west of Greenland"?*

We have made the suggested change.

*L83 "recent, temporally-anomalous GrIS late melt events spanning the end of summer to freeze-up of adjacent ocean waters": Which recent events? Please provide a few examples and references.*

We have slightly edited the text to reflect satellite era studies of late season GrIS melt in late boreal summer and autumn. Two papers referenced include Doyle et al., 2015 and Stroeve et al., 2017.

*L112 and L156: Can you provide some further explanation as to why KAN_B was selected to build T+/T- composites, thus limiting the study period to only 5 years? Especially since the objective is to study melt events, why choose a station on the tundra instead of a station on the glacier? KAN_B has a much shorter data record than all the other K-transect stations, so the selected study period 2011-2015 is only a small sub set of the available data. With the short study period and with composites sub-divided into D_60 and D_30 bins, the D_30 composites comprise only 13 days of data. The next closest station, S5, has data going back to 2003, which would allow for a more robust analysis over a 13-year study period. If you repeat the analysis using S5 to build composites over 2003-2015, do the results shown in Figs. 3-6 and S4-S5 change very much? I think it would be preferable to use station S5 and the longer study period for these figures, or at least mention whether the KAN_B-based results are robust when S5 and the longer period are used. The only place I see a need for a shorter study period is in Table 2 and Fig. 2, where all the K-transect stations are compared over an overlapping period, but if*

*KAN_B were excluded then I think the overlapping period could still extend back to 2008 instead of 2011.*

Our study provides a temporally homogenous examination of above-freezing temperatures along all current PROMICE and IMAU K-transect stations. The study spans 2011-2015 so that both networks of AWS observations can be included. We use above and below freezing air temperatures from Kan_B, located just off the ice edge, as bins for comparing atmospheric conditions about the transect. The intent is not to build the longest record of unseasonal melt conditions, but the most spatially complete.

*L163: It would be helpful to summarize from Table 2 the frequency of T+ events at KAN_B (or S5, if it were used instead of KAN_B to build composites), and mention what percentage of days each represents, i.e., the D_60 bin has 69 events (~46% of days) and the D_30 bin has 13 events (~9% of days).*

We have briefly summarized the frequency of T+ events from table 2 within the first paragraph of the methods.

*L176 "IVT PR is then classified using the SOM technique to produce a matrix of moisture transport patterns, or nodes, that typically occur over the Greenland region": This is very vague. Some additional description should be included for the benefit of readers who are not familiar with SOM - for example, mention that it is an unsupervised machine learning algorithm and that each daily IVT PR field is classified with its closest matching node from the SOM. Are the SOM dimensions and training parameters the same as in Mattingly et al. (2016), or is anything different? It's not clear from this description. It would also be helpful to include in the Supplemental Materials a figure showing the SOM nodes and the wet/dry/neutral groups, so that readers have a better understanding of the SOM classification used in this study.*

We have added some clarifying language in the last paragraph of the methods section on the SOM method and application of the Mattingly et al. (2016) SOM classification, from which we assess frequencies of wet, dry, and neutral IVT patterns as identified by the authors. We refer readers back to the aforementioned manuscript to view the SOM matrix referenced here.

*L178 "As in Mattingly et al. (2016), similar wet (anomalously high), neutral (near climatological median values), and dry (anomalously low) IVT patterns are aggregated, then their frequencies are composited and tested following methods previously summarized": What is meant by "tested" in this context? In the cited paper, the clustering of nodes into wet/dry/neutral groups was subjective, based on visual inspection. I don't recall any testing of this grouping procedure.*

The hypothesis tested is that the frequency of IVT patterns belonging to a given SOM node cluster (i.e. "wet" nodes) differs for T+ events compared to T- events. As you note, the grouping of nodes is based on visual inspection of the IVT patterns for each node, and the grouping procedure itself is not tested in this paper. We have edited the text to clarify this point.

*L208-231 I find it difficult to follow the reasoning in these two paragraphs that leads to the conclusion that "late season melt inferred from T+ events may be driven by synoptic patterns as opposed to local marine forcing" (L234). Why exactly do we reject the hypothesis that local marine forcing drives the T+ events? Is it simply because the K-transect wind directions in T+ events are offshore instead of onshore? What about the other aspects discussed here – such as K-transect wind speeds, sensible and latent heat fluxes over ocean and GrIS, near surface winds in eastern Baffin Bay, etc. – how do they contribute to this argument? These paragraphs should be revised to clarify: if local marine forcing were driving the T+ events, then what would we expect the results here to look like? How do the actual results differ from that?*

We have clarified a hypothesis of what physical dynamic and thermodynamic mechanisms would link Baffin Bay open water to GrIS melt in the last paragraph of the Introduction section.

*L214-218 and Figures S4-S5: Sensible and latent heat fluxes in Figures S4 and S5 are described here as positive in the upward direction. Is this correct? I would have expected sensible heat fluxes over GrIS to be directed from the atmosphere to the ice sheet surface (i.e. negative SHF if the upward direction is positive). In particular, in the K-transect region during T+ events, if air temperature is greater than 0 C but the melting ice surface is at most 0 C, how can SHF be directed upward?*

We have corrected the wording and the positive sensible and latent flux represents downward heat transfer by which the atmosphere heats the ocean and ice sheet.

*L240 "In both T+ cases, low-level winds circulate poleward over north Labrador Sea areas of upward, turbulent heat flux (Figs. S4 and S5), aiding the heat and moisture transfer (as shown by heightened IVT values in T+ relative to T- ) over western Greenland during D60 and D30 (Fig. 5)": Following from my previous comment about the direction of SHF and LHF in Figures S4 and S5, I am wondering if turbulent fluxes over the north Labrador Sea are actually upward and if this region is actually the source of the heat and moisture transported onto western GrIS? Or do the heat and moisture originate from further south, around 30-35S? Subtropical origin seems more likely and is more consistent with the IVT fields shown in Figure 5.*

We have corrected the sign as indicated in the previous comment. While the objective of this work is not to pinpoint origins of subpolar air masses traveling across the ice sheet, we suggest in the Discussions these air masses originate in deeper in the Atlantic than the domain resolved by our synoptic analyses. Upward turbulent heat fluxes (negative sign SHF and LHF in this analysis) over these remote areas may serve enhance the heat and moisture contributions to those air masses.

Of note, we have also changed D30 and D60 to [-30,-1] and [-60,-31], respectively, for consistency between text, figure, and table references to the time periods preceding Baffin Bay DOA.

*L254 "higher 500 GPH values and on-ice lower tropospheric mean winds": The phrasing here seems confusing. "Higher" refers to GPH being higher in T+ relative to T- events, but does "lower" refer to wind speeds in T+ relative to T-, or does it mean "lower troposphere"?*

"Lower" refers to the troposphere and specifically the winds averaged over the 1000-700 hPa levels. We have clarified the corresponding language in the last paragraph of Results section 4.3.

*L267 and Figure S6: Since each station-year is a separate data point, why not include all years of data for each station in Fig. S6 and related discussion? Only using the 2011-2015 period gives a very limited perspective on whether the temperatures are typically correlated or not, since we see statistically significant correlations in only three of five years.*

We prefer to include a temporally homogenous correlation analysis of the nearby oceanic SSTs versus AWS air temperatures for the 2011-2015 period. Subsequent analyses may look to capture longer term local oceanic and coastal/ice sheet temperature linkages as well as the effects of the regional ice sheet-ocean pressure gradient in modulating onshore or offshore flows during T+ and T- periods.

*Typographic Corrections and Figure/Table Formatting*

*Headers for sub-sections 2.2-2.4 should be in bold for consistency with the other headers.*

We have made the suggested change.

*Table 2: I find the labels here a bit confusing. Although each row is labelled as a relative measure (e.g., "S5 vs. KAN_B"), the columns "T+ n[-60, -31]" and "T+ n[-30, -1]" are not relative to KAN_B, they are the actual counts for each individual station.*

We have clarified the language of the table to caption to more clearly describe the table contents.

*Figures 2 and 6: It's difficult to distinguish between the dark and light red/blue colors in the bar charts.*

We have lightened the bars corresponding to the [-30,-1] bin to better distinguish from the [-60,-31] bin in both Figure 2 and 6.

*Figures 3-4: It's difficult to distinguish cyan arrows over blue background. It's also difficult to see the wind vectors for the K-transect stations - perhaps these arrows could use heavier line weights and/or different colors?*

We have made the suggested figure changes.

*Figure 5: Many of the GPH contour labels are cut off at the edges of the figures.*

We have made the suggested figure change.

*L518: The last four references are not in alphabetic order with the rest of the bibliography.*

We have adjusted the placement of references accordingly.

*General comments. The Discussion Paper by Ballinger et al. is an interesting and novel attempt based on in situ, Automatic Weather Station (AWS) observations and Regional-Climate Model (RCM) output to identify the prevailing atmospheric-circulation pattern that causes unseasonal melt events in the southwestern Greenland ice sheet, and whether sea-ice formation and related turbulent-heat production over Baffin Bay in early autumn is of importance.*
*The study is characterized by an accurate and well-structured methodology, but in my opinion, not also interpretations. As an example, the authors conclude that they "find no evidence to support the hypothesis that local open water and resultant turbulent heating has a demonstrable impact on inland ice melt events", even though the approach is purely statistical; the study does not disprove the hypothesis, but rather proves that it is not the dominant mechanism.*

*Structurally, the study suffers from obvious overlooks and requires attention. As examples, I note the allegedly examined AWS pressure and humidity according to the Abstract, even though the analysis is based on AWS air temperature and wind proper- ties only, and the mention of a Sisimiut AWS in the Results and of an Upernavik AWS in the Discussion, without being previously introduced or presenting any related figure in the main body.*

*I recommend a major-revisions status, primarily to provide the authors with ample time to improve the structure and reasoning throughout the text. I also have a comment on the methodology concerning the AWS analysis that, in my opinion, needs to be addressed before publication.*

We appreciate the reviewer's comments on the paper and have addressed their comments below and in the paper accordingly.

*Specific comments*

*The linguistic level is good, however sentences tend to be condensed, and would benefit by a more elaborate style. Since the study is not purely meteorological and aims potentially to appeal to a wider Earth sciences audience, some effort should be invested into familiarizing readers with meteorological norms instead of just mentioning parameters, numbers, and units. Additionally, the reader's knowledge over methodological approaches is taken for granted. As a rule of thumb, a second-year Earth- sciences master's student should be able to follow without referring to textbooks for clarifications.*

Revisions are aimed at structural improvements to improve readability in line with the journal's cryospheric science audience.

*The study has a broadly clear structure, but the argumentation is not straightforward, and complicated – sometimes even speculative – discussion points are widespread in Results and Conclusions. Also, as a rule of thumb, referencing supplementary material in Results or other studies in Conclusions should be avoided, in order for the study to be self-standing.*

*Effort should be invested into describing the theoretical background and the study area, particularly its wind regime and mechanisms, as well as recent Surface Mass-Budget changes within the Introduction or before Data as a self-standing section, so that readers can easily relate afterwards to the Discussion. Recommended literature: Van Angelen et al. (2011), Van As et al. (2014), and the classic GIMEX-90/91 work. The inclusion of a description of the different areas of a glacier, as well as an Equilibrium- Line Altitude definition and how high it is in the study area, is also recommended.*

We have touched upon these components through the manuscript where appropriate to complement given research objectives and details of the K-transect study area.

*The use of KAN_B observations as basis for the analysis needs to be motivated better than just saying that it records more above-zero Celsius air temperatures than all the other K-Transect AWSs. The way I see it, by the end of August, after the end of the melt season, the increased solar-zenith angle and ice-sheet surface albedo limit the influence of solar radiation on potential ice-sheet surface melt, which is thereafter driven by climatic drivers influencing downward longwave radiation (i.e. cloudiness) and turbulent fluxes (e.g. September 2010; Charalampidis et al., 2015). Also, katabatic-wind flow, driven by the balance of downward sensible heat flux and radiative cooling at the surface, becomes gradually colder and denser, and therefore more intense and laminar, thereby hindering turbulent mixing over large part of the AWS transect that is characterized by decreased surface roughness (Smeets and van den Broeke, 2008; see also specific comments for L208-209 and L211-218). Hence, melt events might occur in cases of weak katabatic-wind flow and concurrent northwesterly atmospheric advection from Baffin Bay, or moderate to strong katabatic-wind flow and concurrent southerly barrier winds along the ice-sheet margin driven by synoptic circulation.*

We have clarified our motivation for using KAN_B above (T+) and below (T-) freezing events for compositing across the K-transect, and the compositing methodology, in the Methods section. Acknowledging the ice-albedo feedback is weak in autumn relative to summer months, we have edited the Discussion section to address local, mesoscale, and synoptic winds supporting T+ events. In concluding that synoptic forcing strongly influences our T+ cases, we have also briefly discussed the implications of enhanced moisture advection and resulting cloud longwave effects on melt.

*Located only 1 km away from the ice-sheet margin, KAN B should be the station most sensitive to both these circulation patterns. In both cases, the local boundary layer will become more humid (the role of humidity on energy balance and as a cause of surface melt should be explicitly described), and KAN_B might record positive air temperatures, in which case it is also worth inspecting all on-ice AWSs for positive air temperatures, which would be indicative of surface melt. (It should be explicitly mentioned in the text that positive near-surface air temperature is used as an indicator of melt at the ice-sheet surface.)*

A pan-Greenland evaluation of positive on-ice air temperature anomalies (above-freezing or simply above-normal) is beyond the scope of this paper. Future analyses will look to expand the spatial fingerprint of local and remote forcing of marine intrusions on Greenland air temperature

events, but here we wish to focus solely on the data-rich K-transect for this proof of concept assessment.

*The statistical analysis should reveal which of the aforementioned cases is the most frequent (and not simply differentiate between regional meteorological processes acting during positive and negative KAN_B air-temperature events, as currently stated in L154-155. What goes on during negative KAN_B – and hence K-transect – air temperatures is not the focus per se; it is just used as means of comparison). The above theoretical introduction material along with Figure 2 alone proves, in my opinion, that synoptic circulation in the South of Greenland is the driver of unseasonal melt events mostly in the lower ablation area until the elevation of KAN_L. The persistent katabatic wind regime acts otherwise as a shield against turbulent forcing from Baffin Bay. All the rest should be an elaboration around this key result.*

We have adjusted the wording in the methods section to better focus attention on the common timing and physical processes bringing above-freezing air to KAN_B **and** S5 and KAN_L, thereby generating "unseasonal" melt at latter those on-ice stations.

*It might be insightful to make a similar comparison of KAN_B with observations further away from the ice-sheet margin. I do not know what the result might be, but I am guessing weakening southerly wind away from the margin in case of synoptic-driven, barrier-wind occurrences, and otherwise more intense winds than whatever KAN_B is reporting at that time, directed toward the ice sheet.*

As previously mentioned, follow-up analyses will take a wider spatial examination of thermodynamics and dynamics of unseasonal melt.

*Nevertheless – and this is my major methodological concern – it is not clear in the text how the KAN_B temperature events are defined. Are these definitions based on hourly or daily observations? Is there a time window plus/minus delta_t around a warm/cold KAN_B event t within which the conditions at all other stations are evaluated? Is delta_t selected larger or smaller depending on the duration of the KAN_B event t? Eventually, how sensitive are the statistics, and how conclusive the implications, between hourly and daily analysis? Please, elaborate.*

Air temperature composites are based on daily mean air temperature values above and below-freezing at KAN_B. Therefore, T+ is one bin, and T- represents the other bin. We have taken strides to clarify this partitioning in the manuscript. Duration of T+ and T- events and sub-daily analyses are beyond the scope of the study, but will be examined in follow-up research using climatological seasons.

*Technical comments*
*Please, make use of dashes, or hyphens if appropriate, throughout the text to facilitate readability in case of several nouns in a sequence, e.g. in the Abstract alone*

We have made recommended changes where appropriate throughout the paper.

*L27: air-temperature episodes; L29: open-water duration; L30: sea-ice advance; L31: sea-ice growth; L38: late-season, ablation-zone melt events, but also L129: "Surface\*hyphen\*atmosphere features" Also, differentiate between dashes and minuses.*

We have attempted to use hyphens where appropriate and differentiate dashes and minuses in the text.

*Use the more appropriate "area" instead of "zone" throughout the text.*

We have made the suggested substitutions.

*I recommend the use of "observations" instead of "data", while differentiating between in situ and remotely sensed, and "simulations" for modeling products.*

We prefer to use "data" as a more general reference to avoid confusion as the climate science community often tends to characterize reanalysis output, for instance, as observational or modeled based on variable studied.

*I recommend the use of "average" instead of "mean."*

We prefer to leave the term "mean" as it is used throughout the paper.

*Except for "Buffin DOA", complete "Buffin" as "Buffin Bay" throughout the text.*

We have made the suggested addition of "Baffin Bay" for clarity.

*Capitalize the first letters of every word when an abbreviation is introduced, so that the reader's eye can make a quick connection, i.e. L30: Date Of sea-ice Advance (DOA).*

We prefer to leave the referencing as it is given in the manuscript preceding the acronym.

*L27: delete "significant", and reserve it exclusively for describing correlations.*

We indicate "significant" as it relates to correlations and t-test differences between T+ and T-events.

*L32: "unseasonal melt events. . ."*

We are not clear what this comment refers to, but the phrase has been previously defined in the abstract.

*L34: "Southwest" or "southwestern"*

We have made the suggested change to "southwestern."

*L34: The influence of synoptic and mesoscale systems on the above- and below- freezing near-surface air-temperature events. . .*

We have altered the referenced sentence as indicated in the following point.

*L35: "AWS" -> "the"; "The in situ observations..."; Why are pressure and humidity observations mentioned here? I was unable to spot them later in the study. Why is wind speed and direction not mentioned?*

We have removed the reference to pressure and humidity and inserted some commentary on the AWS wind speed and direction.

*L36: "against" -> "with"*

We have made the suggested change.

*L35-38: I suppose "MAR, RACMO2, and ERA-Interim are used to provide context to the in situ observations by explaining the air-mass origins and the (thermo) dynamic drivers of melt events." As it is written now, it gives the impression that the in situ observations need to be calibrated against RCMs.*

We have altered the wording to clarify the role of MAR, RACMO, and ERA-Interim data. As mentioned, these data provide regional and synoptic context to the local AWS winds. MAR and RACMO have both been extensively calibrated against on-ice AWS observations (e.g. from PROMICE and IMAU station records) as mentioned in the Delhasse et al. (2018) and Noël et al. (2018) references included in the Data section.

*L41: Delete "consistent with previous studies". Try not to refer in the Abstract and Conclusions to other studies.*

We have removed this phrase.

*L42: Consider removing "pressure-gradient driven" from the Abstract, and describe briefly the mechanism in the Introduction.*

We have removed this phrase and paired pressure-gradient discussion with a new composite pressure and 10-m wind figure.

*L42: "Katabatic-wind regime", since the word "katabatic" is an adjective referring to a descending object, in this case, wind.*

We prefer to leave as "katabatic regime."

*L41-42: I think this is an overall complicated explanation. I think the primary obstacle of Baffin-Bay marine-air intrusions over the ice sheet should be intuitively the persistent katabatic wind regime that intensifies in the beginning of autumn. Additionally, barrier winds might be present, in which case melt might occur due to air-mass transport from the Atlantic. I propose restructuring as: ". . .are obstructed by the persistent katabatic- wind regime flowing downslope from the ice sheet, and the occasional occurrence of barrier winds along the ice-sheet margin.", or something along these lines.*

We have made a similar change as suggested to clarify the role of katabatic winds in "blocking" the marine layer transport onto the K-transect.

*L44: "Substantial mass losses. . ."*

We have made the change.

*L46: "have become sensitive" or "are becoming increasingly sensitive"*

We have made the change to the suggested latter phrase.

*L48: Since all Van or Van de Author have been categorized under V in References, consider capitalizing the first letter of "Van" in the in-text occurrences, as well as that of the first author in the References, in order to allow the reader to navigate through easily.*

We elect to leave the author's names as they are given.

*L53: Replace "play a key role" by "are of importance"*

We have made a similar suggested change.

*L54: "surface melt at the southwestern GrIS."*

We elect to more generally include "western" before "GrIS melt."

*L55: "Conflicting evidence have been presented in literature over the past decade regarding. . ."*

We have made the suggested change.

*L56: "Regional-climate simulations. . ."*

We have changed the phrasing to "Regional climate model simulations…"

*L60: "noted that. . ."*

We have made the suggested change.

*L61: Replace "which" by "and"; replace "by" by "to".*

We have made the suggested changes.

*L63: "in Ilulissat and Nuuk, approximately 200 [Unit] to the north and south of Kangerlussuaq, respectively. . ."*

We have added the appropriate reference to km and "respectively" as suggested.

*L66: Replace "coupled" with "correlated".*

We have made the suggested change.
*L67: Please, delete "robust". Statistical quantities are a matter of context, and their interpretation is dependent on the datasets and the research question itself, which in this instance are not present; "(from Markus et al., 2009)"*

We have substituted "robust" for "significant" here.

*L69: "The authors found that significant, positive correlations between Baffin and Labrador SST and coastal SAT often persist. . ."*

We have made the suggested change.

*L70: "onset of freeze" should suffice; "Applying a similar correlative approach on. . ."*

We have adopted the first change, but elect to maintain rather than change the second wording recommendation for this line.

*L71: "while utilizing melt/freeze product. . ."*

We have made a slight change in wording here.

*L74: "Both studies indicated. . ."; replace "upper-level" with "high-altitude".*

We have made the first change, but elect to keep the second as "upper-level" is commonly used to describe winds in the mid-to-upper troposphere.

*L75: Replace "at the limits of" with "toward the end of".*

We have made this change.

*L81: Replace "research studies" with "literature".*

We have made this change.

*L86: Delete "automatic weather stations", since AWS has already been introduced.*

We have made the change since AWS introduced in the Abstract.

*L99: "Daily observations are available over the 1979–2015 period at a. . ."*

We have made this change.

*L101: Delete "day of sea-ice advance", since DOA has already been introduced.*

We have made this change, but have clarified the DOA metric.

*L102: Bliss et al. (in review)*

This work is "in press" and we have updated the reference and in-text citation.

*L112: "in this study (Fig. 1)."*

We have added a the reference to Fig. 1.

*L113: "KAN_B that is situated approximately 1 km away from the ice-sheet margin. . ."*

We have clarified the statement as suggested.

*L117: Delete "tundra (KAN_B only) or glacier ice"*

We have made the suggested change.

*L116-117: The observational distance from the surface decreases during winter season due to the accumulating snowcover around the AWSs by as much as two meters depending on the location along the transect, while it increases again each melt season until the complete ablation of the accumulated snowpack. In the case of KAN_U that is located in the lower accumulation area, this distance tends to become shorter over the course of a few positive mass-budget years due to the incomplete ablation of the snow cover, but also due to the sinking of the AWS tripod during ablation in the temperate firn below, as discussed by Charalampidis et al. (2015). After the extreme 2012 melt season, snow accumulated around KAN_U during the 2013 and 2014 positive mass-budget years, while on 3 May 2015 the half-buried tripod was replaced.*

We appreciate the author's insights on the K-transect AWS air temperature measurement heights, which are complicated by snow accumulation and ablation through the annual cycle. We have added the suggested citation and an acknowledgement that the sensor heights fluctuate with these mass balance components.

*L121: Consider including also Citterio et al. (2015) describing the PROMICE AWS.*

There are many papers describing aspects of the IMAU and PROMICE AWS networks, but we elect to mention a concise list here.

*L125: Delete "have been shown to"*

We have made the suggested change.

*L127: "from between 1000–200 hPa height, corresponding to the distance between sea level and lower stratosphere at 67 N (e.g. Zängl and Hoinka, 2001). . ."*

We appreciate the reviewer's reference, but we elect to keep the original description of integrated vapor transport (IVT), which is elaborated on in the Methods section and follows from work of Mattingly et al. (20016, 2018).

*L129: "Surface\*hyphen\*atmosphere" since the two are related/interacting.*

We have made the suggested change.

*L130: Replace "features" with "interactions". Alternatively, replace "Surface– atmosphere interactions" with "Boundary-layer processes"; 500 hPa height (i.e. ~5000 m above sea level)"*

We have replaced "features" with "interactions" as suggested, but elect to maintain variable descriptions.

*L132: "10-m above surface and 850 hPa height. . ."*

We have shortened the sentence, but elect to keep the core "Wind speed and direction at 10-m and 850 hPa" intact as these represent standard reference levels for these dynamic variables.

*L133: "low-level atmospheric flow. . ."*

We have made the suggested change.

*L143: "ice caps, and improving. . ."*

We have made minor edits to the sentence to make it clearer.

*L145: "to characterize near-surface. . ."*

We have made the suggested change.

*L151: It should be mentioned that above 0 C air temperatures at on-ice AWSs are considered an indicator of ice-sheet melt.*

We have integrated the suggested addition.

*L152-165: It is not clear how the KAN B temperature events are defined. I refer to Specific comments*

We have made efforts to more clearly explain the compositing approach within the Methods section.

*L157: "due to the station's location. . ."*

As with the previous comment, we have clarified the compositing approach.

*L171: "Based on RCM output, . . ."*

We clarify that ERA-Interim data is used for IVT calculation and subsequent classification.

*L172: Delete "integrated water-vapor transport", since IVT has already been introduced.*

We have made this suggested change.

*L183: Refrain from mentioning supplementary material as part of the text, and only refer to them in parentheses at the end of sentences. Nevertheless, supplementary material should be referenced outside Results, and if Fig. S1 should be referenced in the first sentence of the Results, perhaps it belongs in the main article.*

We elect to keep figure references in their current place with the figures included representative of the main results of the manuscript.

*L189-191: Nice, but perhaps also mention that the difference has narrowed primarily due to the prolongation of the melt season over the course of the thirty years.*

We have added some remarks with regards to melt season lengthening.

*L193-195: It is a bit awkward to see all of a sudden a Sisimiut station being mentioned here in the first subsection in Results even though it has not been introduced earlier, and a case being made based on supplementary material. Please, restructure.*

We have introduced the DMI surface air temperature data for Sisimiut and Kangerlussuaq AWS at the conclusion of the Data section. We have also clarified their supplemental usage as they relate to the study.

*L195-199: This belongs in the Discussion.*

We prefer to leave this section in its existing place.

*L201: I am not sure I understand what this first sentence is trying to justify. Also, sounds like it belongs in Methods.*

This justification refers to K-transect's observationally-rich environment, and the use of AWS networks to resolve the spatial fingerprint of subgrid-scale melt.

*L203: Does the term "composite" essentially refer to the average of all instances? Please, clarify.*

We have clarified the term in the Methods section.

*L206: Second half of the sentence refers to T above 0 C events? Please, clarify.*

This refers to differences in T+ and T- events as indicated in the previous sentence.

*L208-209: South-southeasterly winds should have a 157.5 degrees direction, given the orientation of the PROMICE AWSs, so what is now written is incorrect. These winds in autumn are katabatic with a downslope direction (90 degrees) deflected 45 degrees to the right by Coriolis force (i.e. southeasterly direction; cf. Van den Broeke et al., 2009), plus potential southerly synoptic influence. Nevertheless, this quantification seems to be more prominent above the long-term ELA and KAN B. Good thing to also mention.*

We have added some text clarifying the wind direction and Coriolis influence citing the van den Broeke paper provided. We have also mentioned statistically significant changes to wind direction above the long-term ELA at S9 and KAN_U within the paragraph.

*L211-218: This belongs in the Discussion. Nevertheless, I am not sure I agree with this interpretation, and I explain: Strengthening wind speeds during positive air- temperature events at KAN_B do not reflect strengthening of katabatic flow (i.e. cold, dense wind), but rather that additional wind components inducing turbulent mixing might be present. A strengthened katabatic flow would remain cold, and would be more laminar, and hence less turbulent, than usual. Katabatic wind could enhance turbulent mixing during melt season at the lowermost rough-surface parts of the tran- sect, but we are discussing unseasonal melt events, implying that sunlight is reduced, hence surface melt is not sustained in the way it does during the melt season, while surface roughness below ELA might be substantially decreased due to accumulating snow cover, and above ELA slightly increased for the same reason, i.e. sastrugi formation. The observations from most AWSs within the same averaging periods suggest southerly deviation during positive air-temperature events from the cold-event direction, and the way I perceive it, southerly synoptic influence. I note that Figure 2 and Table 2 suggest that melt might occur primarily at the two lowermost on-ice AWSs, while wind speed as well as wind-speed differences between positive and negative cases within the same averaging periods are more pronounced at higher elevations due to comparatively reduced surface roughness (cf. Van den Broeke et al., 2009).*

We have revised our wording here and softened the language. We appreciate the reviewer's feedback and elect to include the present list of factors. As we suggest, these may contribute to

the wind speed intensification and therefore link with turbulent flux analysis that is subsequently presented in the paragraph.

*L214-215: Positive energy flux that contributes to ice melt is most definitely downward, i.e. directed from the atmosphere toward the ice-sheet surface, and not necessarily the result of increased wind, rather the result of increased turbulence; "Sensible heat fluxes" does not sound nice. Consider removing the plural.*

We have made the suggested changes, acknowledging the heat flux sign was incorrectly described.

*L219: The first sentence is not a result. Please, delete or relocate to Introduction.*

We have added "The K-transect" at the sentence outset to provide context to the result.

*L222: "at KAN_M and S9 in the upper ablation area."*

We have made the suggested change.

*L223: Replace "modeled" with "simulated".*

We have made the suggested change.

*L223-225: Difficult sentence, consider revising. Also, please, relocate in the Discussion.*

We have elected to remove part of the sentence that is largely duplicative with text earlier in the section.

*L228-229: This sentence, as it is, belongs in the Discussion. You can reformulate and keep in the Results as: "We note that there is a height difference between the RCM 10-m above-surface output and the measuring heights of the AWSs, as mentioned in the Data section.", or something like this.*

We have altered the sentence and elect to keep it in its present place.

*L233-234: This first sentence is Discussion material. Please, relocate or reformulate, since it sounds like a rather abstract introductory sentence referring to the previously outlined results.*

We have integrated this sentence into the Discussion content.

*L239: Explain the significance of 540 dam (i.e. 5400 geopotential meter), i.e. it often distinguishes solid and liquid precipitation.*

We have added details as requested on the 540 dam contour and included its label in Fig 5 panel plots.

*L255: The link between moist air masses and how they may facilitate ablation-area melt is not clear. Please, elaborate preferably in the Introduction.*

We have added "warm" as is evident when they pass over the GrIS by the increasing height values, tropospheric moisture content, and K-transect surface air temperatures.

*L261: Include a short comment on strong negative NAO phases, as seen in monthly observations.*

We have added the suggested change.

*L266: "turbulent atmospheric heating"*

We have made the suggested change.

*L267-269: Please, refrain from referring to supplementary material as main part of the text, i.e. delete "As shown in Fig.S6". Instead, include a citation at the end of the sentence. It should be mentioned that it is a moderate to weak link that seems to be year-dependent. The way I perceive it, this differentiation amongst years suggests synoptic-circulation control that may or may not be present each year, and that is a good comment to include before anything else. (For example, Fig. S6 suggests weak correlation in autumn 2013 between local sea-surface temperature and AWSs, when limited melt occurred even at KAN_U (September 2013; Charalampidis et al., 2015).*

We adjust the reference to the end of the sentence and restructured the wording. Last in the paragraph we posit that the synoptic control may be stronger in years of higher correlation, and a deeper examination is currently being conducted in related work.

*L275: "statistically robust co-variability": Please, remove "robust" and reformulate.*

We have changed "robust" to "significant" to be consistent with other correlative findings.

*L271-279: This part seems somewhat arbitrary, since no evidence of a different wind regime between 2011–2012 and 2013–2015 was presented. Also, I am not sure I agree with the categorization. Please, revise/clarify.*

We are suggesting some physical linkages, supported by literature, that could explain strong versus weak interannual correlations along an "extended transect" at 67ºN between near-shore SSTs and coastal and on-ice air temperatures. In doing, we address some reasons why the coastal response to sea-ice heat flux feedbacks might be strong, yet very localized. We expand on these concepts further in the following paragraphs of the discussion.

*L287: Mention also the distance from Kangerlussuaq.*

We have added this information.

*L290-294: This is highly unclear and speculative. Please, revise.*

We have revised this area by highlighting that increased ice sheet radiational cooling over the ice sheet interior versus coast, which helps to propagate katabatic winds that block inland penetration of marine air.

*L295: "appears to be of minor importance. . ."*

We have made a minor wording change.

*L304-306: Please, clarify.*

We have clarified the language in the sentence.

*L306: "Denmark Strait and Irminger Sea at the East coast of Greenland. . ."*

We have made a similar change to clarify Denmark Strait and Irminger Sea location with respect to Greenland.

*L316: Generally, Conclusions outline key findings in the Results and key Discussion points. The current state appears more like just another part of the Discussion, while the only Concluding bits are between L319-324 and L341-345. Please, consider rewriting the whole section outlining important quantifications from Results. Please remove all citations, as the Conclusions should refer only to the present study, and should be self-standing.*

We have removed some of the redundant citations, but have kept those in reference to future research avenues that dovetail from our main results.  To that point, we have made an effort to further emphasize the main manuscript results in the Conclusions section.

*L317-319: Consider simplifying; rephrase "around the limits"*

We have changed wording to clarify the limits previously mentioned.

*L377: Delete "2018"*

We have updated the reference.

*Table 1: Include a minus in front of the 1 km of KAN_B, so the reader can immediately see that it is different from the rest of the stations. "50*hyphen*150 m"*

We have described KAN_B's position in the caption.

*Table 2: Were these daily averages? Please, clarify. Note that dashes have been used instead of minuses. Also, make column lines between columns 3-4 and 5-6 thicker to facilitate the eye of the reader.*

We have clarified that the counts involve daily mean air temperatures. We have also thickened the line between the columns as suggested.

*L540: Replace "30 to 60-day window" with "time window defined by the 60th and 30th day before day of ice advance (DOA)"*

We have clarified the caption.

*Figure 2: Include the legend that is shown in panel a also in b and c. In panel c, include also N-S-E-W. In panel d, are these points in reference to daily or hourly values? Please, specify.*

We have included legends for panels b and c, and added North, East, South, and West accordingly to panel c. We have also clarified the "daily mean" values in panel d.

*L551: Define "composites" better.*

As the composite methodology has been clarified in the Methodology section, we wish to keep the captions concise.

*L552-553: Difficult to fathom. Please, clarify.*

We have clarified the caption.

*L553: "selected"*

We wish to keep the wording the same.

*L569: Spell out IVT and include abbreviation since it is used as such in the figure.*

We have made the suggested change.

*Figure 6: Include legend in both panels. In the caption, spell out all abbreviations and include abbreviation in parentheses.*

We have included the legends in both panels and have spelled out abbreviations not previously listed in captions.

*L754: Define "composites" better.*

As the composite methodology has been clarified in the Methodology section, we wish to keep the captions concise.

*L577-578: Difficult to fathom. Please, clarify.*

We have clarified the caption in a similar manner to the Figure 2 caption (L552-553).

*Table S1: Dashes as minuses. Please, correct. "2011\*hyphen\*2015"*

We have made the suggested change.

*Figure S2, S3, S4, S5: Spell out DOA initially, and then use abbreviation.*

We have spelled out date of sea ice advance (DOA) in all captions after the terms initially appear.

*Figure S4, S5: Define composites better.*

As the composite methodology has been clarified in the Methodology section, we wish to keep the captions concise.

*Figure S6: Introduce SST properly in caption.*

We have spelled out SST as suggested.

References
Charalampidis, C., van As, D., Box, J. E., van den Broeke, M. R., Colgan, W. T., Doyle, S. H., Hubbard, A. L., MacFerrin, M., Machguth, H., and Smeets, C. J. P. P.: Changing surface–atmosphere energy exchange and refreezing capacity of the lower accumulation area, West Greenland, The Cryosphere, 9, 2163-2181, doi:10.5194/tc-9-2163- 2015, 2015.

Citterio, M., van As, D., Ahlstrøm, A. P., Andersen, M. L., Andersen, S. B., Box, J. E., Charalampidis, C., Colgan, W. T., Fausto, R. S., Nielsen, S., and Veicherts, M.: Automatic weather stations for basic and applied glaciological research, Geol. Surv. Denmark Greenland Bull., 33, 69–72, 2015.

Van Angelen, J. H., van den Broeke, M. R., and van de Berg, W. J.: Momentum budget of the atmospheric boundary layer over the Greenland ice sheet and its surrounding seas, J. Geophys. Res.- Atmos., 116, D10101, doi:10.1029/2010JD015485, 2011.

Van As, D., Fausto, R. S., Steffen, K., and the PROMICE project team: Katabatic winds and piteraq storms: observations from the Greenland ice sheet, Geol. Surv. Denmark Greenland Bull., 31, 83–86, 2014.

Zängl, G., and Hoinka, K. P.: The Tropopause in the Polar Regions, J. Climate, 14, 3117–3139, doi: 10.1175/1520-0442(2001)014<3117:TTITPR>2.0.CO;2, 2001.

3. Anonymous Reviewer #3

*General comments:*

*Ballinger, et al propose a hypothesis for the onset of late-season melt events on the Western Greenland ice sheet. Some prior literature has suggested that open-water ice- free conditions in Baffin Bay cause melt incursions onto the ice sheet, but Ballinger, et al use weather stations data, regional climate models and reanalysis products to make a case that incursions of North Atlantic warm air are responsible for these late-season events, regardless of sea-ice conditions.*

*Overall, the hypothesis is novel, and I applaud the authors for bringing in multiple datasets to support it. As one of the other reviewers also noted, I am, however, very concerned that the short time series (2011-2015) limits the conclusions that can be made with any statistical certainty. From the evidence given, I would say the study is suggestive, but not conclusive in its arguments. In my opinion, the authors have a choice: either use a longer time-series of measurements or model data (for instance, keep the 2011-15 analysis of AWS data but make the regional climate data analysis span from 1979-2015, which is possible given that the two datasets aren't even used in direct comparison to each other), \*or\* soften the conclusions to make it clear they are suggestive but not entirely conclusive given the short time series. With only 5 years at your disposal, random noise can easily be interpreted as interesting new patterns.*

*As always, I acknowledge that the authors have spent more time considering this paper than reviewers spend reading it. If authors believe my judgements are unfounded or based on a misunderstanding, or if I just missed something in a comment, they may of course make that case upon resubmission to the reviewers or editor. Overall I do think the study, and the hypothesis put forward, is quite suggestive and worth publishing for that reason, even with the concerns cited, but the concerns should be addressed.*

We thank the reviewer for constructive comments and have addressed their concerns in an attempt to improve the quality of the manuscript.

*Specific comments:*

*L31-32: "For the unseasonal melt period preceding the DOA" . . . this make it sounds as if it's referring to a \*specific\* melt event that hasn't yet been mentioned. Generalizing the sentence more to something like "For periods of unseasonal melt preceding the DOA,. . ." may be clearer to the reader.*

We have clarified the wording as suggested.

*L34-35: "the above and below freezing surface air temperature events. . ." same comment as above.*

We have made a similar suggested change.

*L83-84: "temporally-anomalous GrIS late melt events" : in much of the text, terms such as "unseasonal" and "temporally-anomalous" are used, but never really defined. Does is refer to melt events after a certain date/season? Recent late-season melt events that are more frequent than previous periods? The descriptor is a bit vague here.*

We have clarified the language, relating late-season melt (as in the manuscript title) to "unseasonal" so as show the reader the terms are used interchangeably.

*L104: (quick format check) Several headings are not "bolded", while the rest are. Quick fixes.*

We have made bolded headings for consistency.

*L119: "erroneously low values are filtered out prior to analyses" : Which variables were filtered in this way: temperature, wind speed, others? If you can, be more specific.*

We have clarified the which variables were filtered.

*L152: "A composite approach is applied to characterize atmospheric conditions. . ." : From what I saw, I couldn't find an explanation in the text of how these composites were put together, or what exactly the reader is looking at when seeing the composite numbers in the tables and figures. I don't expect the authors to provide a textbook lesson on composite statistics, but a brief 1-3 sentence explainer for the reader would help reproducibility and clarity of the text. Right now, if I took the same data and tried to reproduce the results I would have no guidance of how to perform the composite analysis other than the authors' word that they applied a "composite analysis." More specificity is needed here to make the results reproducible. If the method is identical to one used in other papers, a citation may suffice.*

We have clarified the composite approach in the Methodology section so that results could be reproduced for additional transects about the GrIS, for instance.

*L162-165: The description of the timespans used in the composite approach is good, I would put this at the top of the paragraph before describing why composite approaches were used and what constraints were placed upon them.*

We have re-arranged the composite descriptions to introduce the temporal restraints before discussing the temperature bins. Care has also been taken to clarify the methodology as mentioned in the previous comment and following the recommendations of the other reviewers.

*L183: "Perspectives on... are shown" is a vague term, almost meaningless. If the sentence is meant to point out something specific that Figure S1 shows, describe it explicitly.*

We have edited the wording to be more direct in what Fig. S1 is presenting.

*L183-199: The first several paragraphs of the Results refer to figures that are almost exclusively in the supplement (Figs S1-S3, with exception of the Fig. 1 map), which makes it awkward for the reader to follow along. These figures appear to be central to the results, not just supplementary. Unless you are limited by the number of figures, consider putting some of the these in the central text rather than forcing the reader to flip back and forth to an entirely separate document just to follow along with your argument.*

We prefer to include figures corresponding to the main manuscript results (i.e. local and regional drivers of K-transect melt events) as they are currently placed, and leave the introductory results on GrIS and Baffin Bay end of melt season overviews, for example, as Supplementary Material.

*L192-193: "Relative to the climatology. . . date in 2012-2015 (Table S1)." It is unclear why 2011 is separated from 2012-2015 here, or why this sentence exists at all. Inter-annual variability has always been present, and no evidence is provided that this juxta- position of years shows any more significant differences than any other random 5-year period in Greenland's climatological history (perhaps it is, but it isn't demonstrated). If not, just omit it.*

As shown in Table S1, the date of Baffin sea ice advance (DOA) in 2011 was the only year of the five years studied with anomalously early ice formation. We wish to briefly note this anomaly as we describe the late melt season characteristics setting the stage for the core results of the paper.

*L195-199: "Interannual differences. . . found in the east and north (Curry et al., 2014)." These sentences don't show new results, but discuss the context of other literature. Consider moving to Discussion.*

We elect to keep these results in their original place to provide context on the ice melt season preceding the primary results of the paper.

*L201-202: "The spatial coherence of observations across the K-transect along with inhomogeneous GrIS Region 3 spatial melt patterns and satellite pixel contamination issues at the tundra-ice interface, lead us to assess the melt events at the station level." It's very unclear what is meant by "inhomogeneous GrIS Region 3 spatial melt pat- terns", or why that would motivate a K-transect station-level approach. Please clarify. Similarly with "the spatial coherence of observations across the K-transect".*

In the vast GrIS region (~539,375 km$^2$) melt is not spatially, nor temporally, consistent. For instance, on the same early autumn day the southern region may experience a strong warming signal, but the northern-most bounded area may not. Due to this inconsistency and issues classify melt for pixels intersecting the land surface, we focus on a station-based approach to identifying on-ice melt events. We have further clarified the language in results section 4.2 to further reflect our justification as mentioned.

*L208-209: ". . .and comparatively becomes slightly more southerly"* One issue with the way these composite records are presented is that there are no uncertainties or spreads presented with them at all, other than which ones are/aren't statistically significant. Thus, comparisons between them, such as "slightly more southerly" are impossible to make without knowing whether or not the difference are simply within the noise of the two datasets, or are just part of statistical noise. This is the case with all the comparisons, actually, and it makes the conclusions difficult to defend.

We acknowledge that there are some shortcomings of the composite methodology as mentioned in your comment, and our comparisons (by time window and temperature threshold) are the main comparisons we explore through the paper.

*L223: "Modeled wind speeds are more intense during T+ versus T- events"* How much more intense? 2 %? 150 %? As noted in the comment above, is the difference greater than statistical noise?

We have noted that simulated winds during T+ versus T- events are ~20-50% stronger consistent with the differences in the AWS observations.

*L227: "with low root mean squared errors (not shown)."* Unless I'm misunderstanding this statement, RSMEs for two values should just be two numbers. They can be stated explicitly instead of just saying (not shown). If they weren't computed, this statement should be omitted.

We have summarized the RMSE for the model and observational comparisons mentioned.

*L227: A slight positive bias in both models is evident. . ."* I assume a slight positive bias in wind speed? Clarify so that the sentence can stand on its own without ambiguity.

We have clarified the wording here.

*L235: The "North Atlantic region" is not explicitly defined here, it is just vaguely referenced even though the authors are clearly looking at particular portions of the map. This ambiguity makes the rest of the entire paragraph extremely difficult to decipher from a reader's perspective, given that different portions of "the North Atlantic" behave in different manners in Figure 5.*

We have clarified our reference area by mentioning Greenland as the geographic focus within the Northwest Atlantic region.

*L235-238: "Whereas T- events "left panels" tend to be characterized by northerly winds over the 1000-700 hPa layer, . . ."* It is extremely unclear what is being referred to here. Are these northerly winds specifically in Baffin Bay, or over the whole figure, or. . .?? Also, this sentence is disjointed and somewhat convoluted, its meaning unclear to the reader.

The "panels" refer to Figure 5 and to clarify we direct the reader back to these in the text.

*L238: "found over ice sheet" –> "found over the ice sheet"*

We have made the suggested change.

*L238-239: "with the 540 dam contour" The 540 dam contour is very unclear in most panels of figure 5. Consider making it clear to the reader exactly which contour is 540 without them needing to interpolate between other lines, if you're going to make a point specifically about the 540 line. Make it easy on your readers to see your point. (Also, as another reviewer noted, make contour labels fully visible, not partially cropped by the panel edge.)*

We have made all contour lines visible and labeled the 540 line so that it is easier for the reader to compare its position between T+ and T- events.

*L246-255: I found this paragraph particularly good and compelling.*

Thank you for your comment.

*L265: "toward earlier (later) melt (freeze)" I understand this sentence construct, and occasionally it's useful, but it's also extremely awkward to read and should only be used when necessary to save space. In this case, "toward earlier melt and later freeze" says the same thing with only one more word, and is much clearer English for the reader to understand.*

We have made the suggested wording change.

*L265: "This hotspot of melt" It's unclear if you're referring to a particular hotspot of melt referred to earlier (if so, point it out), or making a more general statement here.*

We have generalized the wording as suggested.

*L267-269: "Sisimiut SSTs fluctuate. . . ablation zone at S9." It's unclear why this sentence is important. Additionally, it's unclear that a statistically significant difference can be inferred at all if it only happened in 3 of 5 years, but not the other two. Even with just 5 years of data it's hard to make firm conclusions about climate patterns. . . moreso if just picking three years selectively out of those five. This needs to be better justified, or omitted completely.*

This sentence references correlations between local SSTs and air temperatures extending from the coastal to lower accumulation zone at KAN_U.  S9 marks the location from which the correlations in most years drop off and are generally weaker toward the interior.  We have clarified the wording in the paragraph to reflect this statistical finding.

*L271-279: The two points made at the end of this paragraph seem difficult to support conclusively with only 5 years at your disposal, comparing the first two years to the second three. The problem with such a short time series is that signals can be easily interpreted from random noise, making such conclusions problematic at best. This is emblematic of the greatest weakness of the whole paper, making strong conclusions about climate patterns from only five years of data. It is unclear that if you had a 30- year record (long enough to infer at least some of the variability of the patterns you describe), that the same inferences could be made in any significant way.*

We acknowledge that five years of K-transect case studies provide a limited temporal perspective and therefore must be considered in light of the evidence presented that Baffin open water and onshore winds do not appear to be a primary factor driving late-season melt events. However, the study provides a framework to be expanded upon in future, related studies presuming the on-ice AWS programs continue for years to come.

*L289: "near-surface air penetrates at least to" –> "near-surface air often penetrates at least to" (There isn't any indication that it always does.)*

We have made the suggested change.

*L300-301: "positive (negative) GBI (NAO) values" –> "positive GBI and negative NAO values" (more readable)*

We have made the suggested change.

*L332-334: It is good that the authors recognize that a longer time series would help with these analyses. It is still unclear that all of the conclusions in the paper can be made so confidently with the short time series available, and that some of the results are more just "suggestive" than "conclusive."*

As mentioned above, we acknowledge that five years of analysis precludes overconfidence in the results.  However, the study provides a framework to further explore unseasonal GrIS melt characteristics amidst ongoing expansion of the Arctic melt season.

---

## Author Response (AR2)

**Greenland Ice Sheet late-season melt: Investigating multi-scale drivers of K-transect events**

Thomas J. Ballinger, Thomas L. Mote, Kyle Mattingly, Angela C. Bliss, Edward Hanna, Dirk van As, Melissa Prieto, Saeideh Gharehchahi, Xavier Fettweis, Brice Noël, Paul C.J.P. Smeets, Carleen H. Reijmer, Mads H. Ribergaard, and John Cappelen

Manuscript submitted on 19 December 2018, revised on 10 April 2019 & 15 July 2019

Editor's Comments

Your revised manuscript was sent for another round of reviews (to the same referees who reviewed your first draft) as the referees suggested major revisions. I've now heard back from two of the referees (see the referee's comments below). Based on these comments, mainly from the reviewer #1, I ask you to carefully revise your manuscript before I can consider it for publication. Please note that TC journal cares about the conventional structure of scientific papers and authors are discouraged from moving the bulk of key figures to the supplementary material, as well as from referencing the \*whole\* methodology to previously published papers. I encourage you to add more substance and to improve the clarity of material presented in the Methods section, despite this resulting in additional few pages to your manuscript.

Authors' Statement

We thank the editor and reviewers (whose comments are addressed below) for the thoughtful and constructive comments on our revised manuscript. We have considered and subsequently addressed concerns and we believe revisions made on the current version have strengthened the paper, and hope improvements have now rendered the manuscript suitable for publication.

To summarize, the editor's main concerns were that key figures to the manuscript's story needed to be integrated into the manuscript and the methodology needed to more explicit and detailed in describing the techniques utilized in the study. In response to the concern over key figure placement, we have integrated the majority of the figures into the main text of the paper to smooth the reading process. In response to the methodological concern, we have expanded our description of both the integrated vapor transport (IVT) metric and the self-organizing map classification of IVT (please see Section 3, i.e. Methods)

The series of minor revisions by Anonymous Referee #2 have also been addressed. In carefully considering and addressing these areas, we believe the manuscript is greatly improved from the first revision. In the responses that follow, reviewer comments are shown in *italics* while author responses are provided in the red color used here.

**Report #1/Anonymous Referee #2**

*Review of the revised manuscript of Ballinger et al., The Cryosphere Discussions, April 2019*

*In view of the online discussion, the manuscript maintains an unorthodox structure. There are good reasons why there are strict boundaries among different sections in scientific writing, and I am sure the experienced authoring team agrees. The Results section is still largely based on supplementary material, which is at the very least inconsiderate toward the reader. These aspects remain serious disadvantages of a study that begins sprinting and purposely trips itself.*

*As far as the science is concerned, the manuscript appears improved. Apparent mistakes and wrong statements previously evident in the manuscript have been corrected. The study now adopts moderate language appropriate for statistical analyses.*

We appreciate the reviewer re-reviewing our paper.  To improve flow and readability, we have integrated most of the supplemental material into the body of the revised manuscript.

*Specific comments:*

*L115-117: There is logical inconsistency in this justification: KAN_B plays a central role in the study, so perhaps describe what is so special about KAN_B, why it is considered as reference, and then explain how its operational timespan limits the study period. Also, is KAN_B actually uninfluenced by flow patterns through the local topography?..*

We have clarified the importance of KAN_B in Section 2.2, indicating that its location is nearest the ocean (relative to the other K-transect AWS sites) and proximate location to the terminus makes it a logical location from which to composite air temperatures.  We also mention that inclusion of KAN_B allows us to maximize the longitudinal (west-to-east) extent from which to assess possible maritime influences on K-transect temperature conditions.

Regarding the large-scale atmospheric circulation, while the near-surface KAN_B winds are weaker relative to the higher elevation K-transect AWS sites (Fig 2b), the MAR and RACMO winds suggest the low-level flow, influenced by synoptic and katabatic regimes, is roughly consistent from KAN_B to KAN_M (see T+ composite maps in **Fig 7** and **Fig 8** in the revised manuscript).  Further assessments of large-scale circulation attribution to T+ and T- events are left for future work.

*L124: How can there be "erroneously low" daily temperature and wind averages? Are there also erroneously high daily averages?*

We have removed reference to erroneous extremes as these were unintentionally leftover from humidity analyses removed from the manuscript.  The sentence has been clarified to reflect that missing values were removed, and not estimated, before analyses were undertaken.

*L160: Specify that it is above-freezing daily Tair. At the end of the sentence, cite a suitable, positive degree-day modeling study and/or Hock (2005).*

We have added the Hock (2005) citation as recommended.

*L165-166: Is sea-ice positively correlated with Tair?*

We have restructured the sentence to more clearly reflect the correlations between Tair and sea ice freeze onset and SSTs noted in Hanna et al (2009) and Ballinger et al (2018).

L168-170: Unclear to me.

We have revised the sentence to more clearly express our composite methodology.

*L199: End of the Northwest Atlantic melt season area: What is this in reference to? Very unclear.*

We have changed the wording here to be more clear and specific, and have included the figure mentioned in the sentence. The sentence now reads " Time series depicting the conclusion of melt conditions across Baffin Bay and the western GrIS are shown in **Fig. 2**."

*L199: Upward?..*

We have removed "upward" and made the sentence more concise by beginning it with "Later sea ice formation…

*L208: on average 6 days later*

We have made the suggested change.

*L217-220: This sentence is redundant, at least at this point.*

We have removed this sentence.

*L220: I am still missing why KAN_B is so special.*

In the revised description of the K-transect AWS data (Section 2.2) that is motivated by the L115-117 comments previously addressed, we have clarified language justifying the use of KAN_B for analyses throughout the paper.

*L225-227: I would begin with KAN_B, which receives stronger, more southerly winds during T+ events that tend to become weaker and more southerly in autumn. There is general wind-speed increase during T+ events, while direction remains more or less unaltered. Statistically significant directional change is always more southerly, which seems to be more evident above long-term ELA.*

We have clarified this paragraph as follows: "KAN_B receives stronger, more southerly winds during T+ events that tend to become weaker and more southerly in autumn (i.e. [-30,-1]).

There tends to be a general wind speed increase during T+ events, while direction remains more or less unaltered. Statistically significant directional change is more southerly, which is evident above the long-term equilibrium line altitude at S9."

*L232: Open water persistence is unclear to me.*

We have changed the phrasing, which now reads "longer periods of nearshore open water."

*L233: Define MSLP in the text.*

We have spelled out the acronym.

*L245: RMSE = ... [units]*

We have added the correct unit in reference to the wind speed ($ms^{-1}$).

*L248: Why except? I don't see it.*

We have removed "Except during [-30,-1] T+ events" and clarified references to the low-level wind composites presented in **Figs. 7 and 8**.

*L263-264: It already becomes apparent from Fig. 5 that the furthest in autumn, the higher IVT is necessary for T+ events to occur. Good thing to mention here before analyzing the last figure.*

This is a good point to highlight, and we have changed the sentence wording to reflect that higher IVT not only produces melt events, but greater vapor quantities are needed to produce melt as autumn progresses. This point is also now emphasized in the Abstract.

*L270: Change enhance with cause?..*

We have the suggested change.

*L290: Fig. S7?..*

We have corrected the figure number upon integrated most of the supplemental material. The referenced figure is now **Fig S1**.

*L295: The word forcing is redundant.*

We have removed "forcing" as recommended.

*L298: Change manuscript with study.*

We have made the suggested change.

**Report #2/Anonymous Referee #3**

*Overall, I am happy with the revisions made. The authors acknowledge the inherent limitations of making temporal conclusions solely based upon a 5-year time period, but have adjusted the text to reflect that. With the short time period involved, it is (in my opinion) still difficult to make strong conclusions about temporal atmospheric changes, however the results are very suggestive and very worthy of further exploration (which the authors have expressed interest in pursuing). The hypotheses are presented in a way that encourages other groups to test or build upon them with other studies. I think this is a good contribution to the science, the concerns I had in the original manuscript have been addressed, and any particular critiques from me at this point could be categorized as "nitpicking." Thus, I recommend publishing the revised manuscript as is, assuming the other referees are as satisfied.*

We appreciate the reviewer taking the time to review our revised paper.

[revised manuscript text omitted]